# FUAS-Agents: Autonomous Multi-Modal LLM Agents for Treatment Planning in Focused Ultrasound Ablation Surgery

## Abstract

Focused Ultrasound Ablation Surgery (FUAS) has emerged as a promising non-invasive therapeutic modality, valued for its safety and precision. Nevertheless, its clinical implementation entails intricate tasks such as multimodal image interpretation, personalized dose planning, and real-time intraoperative decision-making processes that demand intelligent assistance to improve efficiency and reliability. We introduce FUAS-Agents, an autonomous agent system that leverages the multimodal understanding and tool-using capabilities of large language models (LLMs). By integrating patient profiles and MRI data, FUAS-Agents orchestrates a suite of specialized medical AI tools, including segmentation, treatment dose prediction, and clinical guideline retrieval, to generate personalized treatment plans comprising MRI image, dose parameters, and therapeutic strategies. The system also incorporates an internal quality control and reflection mechanism, ensuring consistency and robustness of the outputs. We evaluate the system in a uterine fibroid treatment scenario. Human assessment by four senior FUAS experts indicates that 82.5%, 82.5%, 87.5%, and 97.5% of the generated plans were rated 4 or above (on a 5-point scale) in terms of completeness, accuracy, fluency, and clinical compliance, respectively. In addition, we have conducted ablation studies to systematically examine the contribution of each component to the overall performance. These results demonstrate the potential of LLM-driven agents in enhancing decision-making across complex clinical workflows, and exemplify a translational paradigm that combines general-purpose models with specialized expert systems to solve practical challenges in vertical healthcare domains. Our code is available at: https://anonymous.4open.science/r/FUAS-7D56

## 1 Introduction

Focused ultrasound ablation surgery (FUAS) is a non-invasive technique that uses focused ultrasound to induce coagulative necrosis in targeted tissues through thermal and cavitation effects. Clinical evidence supports its efficacy in treating a wide range of benign and malignant solid tumors, including uterine fibroids, hepatocellular carcinoma, pancreatic and prostate cancers, and breast fibromas, as well as certain non-neoplastic diseases CMA EC-CTSFUS (2020). Compared to conventional surgery, FUAS offers advantages such as reduced trauma, faster recovery, and fewer complications Liu et al. (2021), making it increasingly attractive to clinicians and patients. However, challenges remain, including limitations in multimodal image analysis, reliance on operator experience for dose determination, and the lack of individualized treatment planning Zhou and Yufeng (2017), which hinder its broader clinical adoption and application.

Previous studies have integrated AI techniques into FUAS to address key challenges in clinical workflows. For image processing, Sun and Zhang (2018) combined GCN with the DMAC model for automatic preoperative lesion segmentation. Zhang et al. (2020) proposed a 3D CNN-based framework for non-rigid MRI-ultrasound registration using unsupervised learning. Intraoperative monitoring using deep learning has been explored by Slotman et al. (2023) and Ning et al. (2020) to support clinical decisions. For dose prediction, models like MLP and XGBoost estimate therapeutic parameters Hu et al. (2023), while others use Deep Multimodal Teacher-Student (MMTS) frameworks to predict thermal doses from ultrasound signals Luan et al. (2024). Machine learning models

have also predicted non-perfusion volume reduction and tissue regeneration after FUAS Zhang et al. (2022).

Despite these advancements, AI-based approaches in FUAS—such as image segmentation, dose prediction, and intraoperative monitoring—often rely on task-specific, expert-driven models. These models face challenges in generalization, dependency on annotated data, and poor adaptability, limiting FUAS technology's broader adoption. This highlights the need for a unified framework with multimodal semantic understanding, autonomous reasoning, and cross-task generalization to advance automated and personalized treatment planning Zhang et al. (2025); AlSaad et al. (2024).

In recent years, multi-modal large language models (MM-LLMs) and agent-based technologies have gained traction in healthcare Thirunavukarasu et al. (2023). These technologies, leveraging capabilities in natural language understanding, multimodal integration, and autonomous reasoning Raiaan et al. (2024) Han et al. (2024) Yao et al. (2025), position LLM-based medical agents as key enablers to address real-world healthcare challenges Vrdoljak and Boban (2025) Yang et al. (2023).

To address practical challenges in the clinical application of FUAS, this study develops a FUAS-Agents system, integrates medical image processing, radiomics, and machine learning to perform MRI segmentation, predict treatment dosage, and generate surgical plans (Figure 1). The system is designed as a MM-LLMs based multi-agents framework, where different agents specialize in tasks such as plan, tool use, memory and optimize. The key contributions are as follows: (1) We propose an AI agent framework for automated FUAS treatment planning, highlighting the potential of agent-based technologies in concrete clinical scenarios; (2) By integrating foundation models with domain-specific expert models, we formulate a hybrid architecture tailored to healthcare, providing a scalable and generalizable framework for agent-driven clinical problem-solving; (3) The system optimizes the FUAS treatment process, supports clinical decision-making, and enhances the precision and personalization of therapy, contributing to the broader adoption and development of this technique.

## 2 RELATED WORK

### 2.1 MULTI-MODAL LARGE LANGUAGE MODELS IN HEALTHCARE AND MEDICINE

With the advancement of LLMs in natural language processing, their application in medicine and healthcare has rapidly expanded. Early models such as BioBERT Lee et al. (2019); Alsentzer et al. (2019) and ClinicalBERT, pre-trained on large-scale biomedical corpora, have significantly enhanced performance in clinical, educational, and research tasksvThirunavukarasu et al. (2023). More recently, domain-specific LLMs like GatorTron Peng et al. (2023), PubMedGPT, and Med-PaLM Singhal et al. (2023) have demonstrated superior capabilities in medical text understanding and generation, particularly in electronic health records (EHRs), clinical question answering, disease prediction—showing increasing potential for interactive reasoning with clinicians. Nevertheless, their unimodal nature limits performance in real-world clinical settings characterized by complex, heterogeneous data. To address this, researchers have shifted toward multimodal foundation models, such as visual-language models. Cross-modal pretraining frameworks like CLIP have enabled image–text alignment and facilitated the development of medical multimodal models (e.g., BioMedCLIP, LLaVA-Med) Radford et al. (2021); Li et al. (2023) , advancing applications in diagnosis, visual question answering, and automated report generation Hartsock and Rasool (2024).

### 2.2 AI AGENTS GENERATE MEDICAL PLANNING

Advances in LLMs have shown AI agents' promise in healthcare, including clinical decision support, disease diagnosis, and hospital management Kim et al. (2024b); Ferber et al. (2024). However, their role in treatment planning remains underexplored. Early efforts, like cGAN-based agents for radiotherapy planning Li et al., lacked foundation model integration, limiting their ability to process complex language and multimodal data Dong et al. (2024).

With advances in LLMs, AI agents have shown promise in healthcare applications such as clinical decision support, disease diagnosis, medical report generation, and hospital management Kim et al. (2024b); Ferber et al. (2024). However, their use in treatment planning remains limited. Early efforts, such as cGAN-based agents for head and neck radiotherapy planning Li et al., lacked integra-

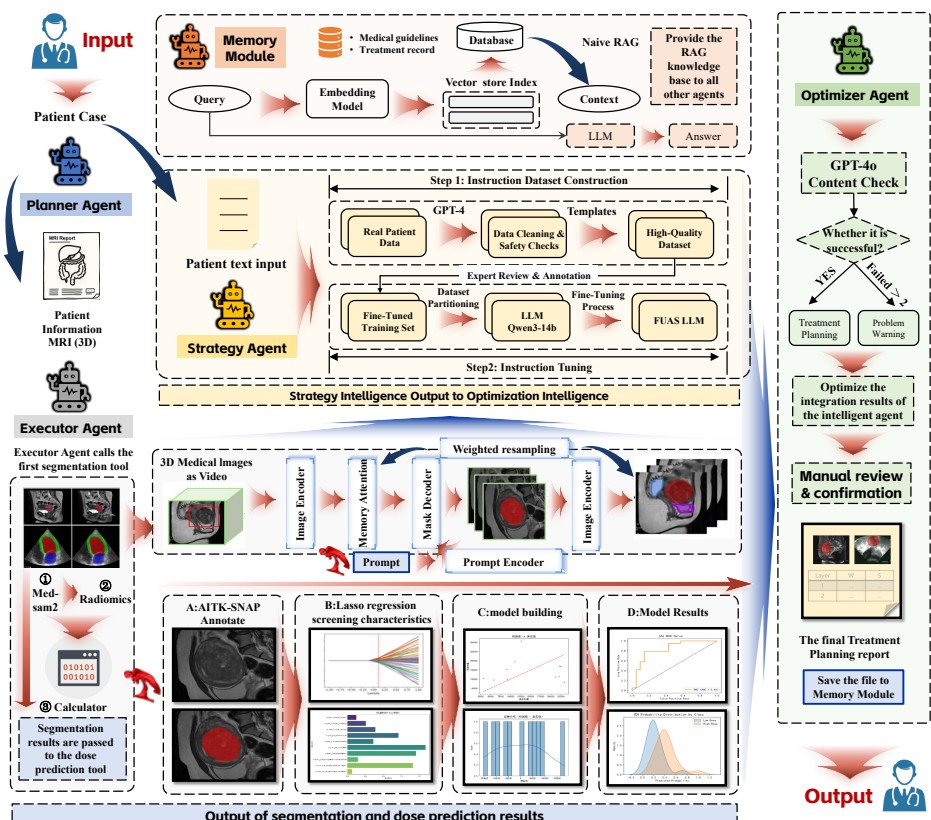

Figure 1: System overview of FUAS-Agents. The Planner coordinates inputs, the Executor performs segmentation and dose prediction, the strategy generates treatment plans with knowledge support, and the optimizer refines outputs.

tion with foundation models, restricting their ability to process complex language and multimodal data Dong et al. (2024).

Recent systems like DOLA Nusrat et al. (2025) have incorporated foundation models, RAG, and reinforcement learning to improve radiotherapy planning. DOLA simulates collaboration between dosimetrists and physicists for automated planning, while Harvard's TxAgent Gao et al. (2025) integrates medical tools for individualized treatment recommendations, focusing on drug interactions. The Medical World Model (MeWM) Yang et al. (2025)uses a closed-loop decision-making framework for tumor treatment planning through observation, simulation, and strategy optimization.

Despite advancements, AI-driven treatment planning remains challenging due to the need for robust multimodal data fusion, clinical reasoning, and personalized adaptation Hsu et al. (2025). Developing AI agents for multimodal treatment planning is a promising research direction.

Compared with these research, FUAS-Agents differs in three key aspects. First, it implements an end-to-end clinical processing simulation tailored to a specific interventional procedure (FUAS), integrating segmentation, dose prediction, strategy generation, and optimization within a unified multi-agent architecture. Second, it performs multimodal fusion across images, clinical variables, structured reports, and domain knowledge, rather than relying solely on language-based reasoning or single-modality vision cues. Third, FUAS-Agents offers a practical paradigm for vertical-domain medical AI, showing that reliable clinical-grade systems require more than general-purpose LLMs and prompt engineering: effective solutions must couple foundation models with domain knowledge, specialized tools, and task-specific fine-tuning.

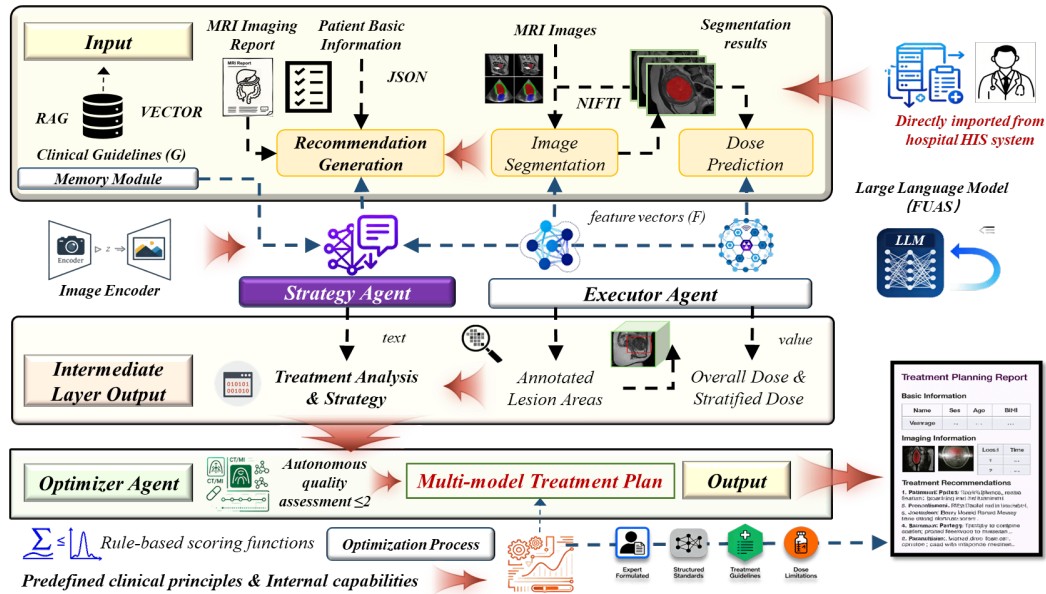

Figure 2: **Overview of FUAS-Agents' data flow and data format.**

# 3 METHODOLOGY

FUAS-Agents is a modular multi-agent system designed for personalized treatment planning in Focused Ultrasound Ablation Surgery. Inspired by clinical processes, it simulates the physician's reasoning, enabling image interpretation, dose estimation, and treatment strategy development through collaborative and knowledge-based interaction. FUAS-Agents follows an "agent-tool-memory" architecture, consisting of: (1) Agents, responsible for reasoning, planning, and validation; (2) Tool Modules, which provide deterministic capabilities like image segmentation and dose prediction; (3) Memory Module, a shared knowledge base accessed by all agents via a RAG interface, storing clinical guidelines and prior cases. A detailed workflow of the FUAS-agent, focusing on the data flow and data formats, is presented in Figure 2.

The Planner Agent decomposes tasks and coordinates downstream modules. The Executor Agent, upon receiving instructions, selects tools. Segmentation and Dose Prediction Tools extract imaging and quantitative parameters, which are passed to the Strategy Agent to generate a patient-specific treatment plan. The Optimizer Agent conducts symbolic validation, triggering a reflection mechanism for plan revision if inconsistencies arise. All agents access the shared knowledge base, supporting the Strategy Agent in generating consistent plans and the Optimizer Agent in plan refinement.

## 3.1 PLANNER AGENT

The Planner Agent is the entry module of FUAS-Agents, responsible for task orchestration and coordination. It interprets clinicians' instructions, organizes patient data, and decomposes treatment planning into subtasks. Inputs include multimodal information and clinician requirements, while outputs are structured task plans defining execution order, tools, and data formats. The Planner Agent collaborates with the Executor Agent for segmentation and dose prediction, and with the Strategy Agent for treatment generation, thus ensuring logical consistency and clinical relevance at the process outset.In the subsequent text, the symbols for the formulas and the results of the data input and output are shown in Appendix A.

## 3.2 EXECUTOR AGENT

The Executor Agent is the core execution module of FUAS-Agents, responsible for invoking tools assigned by the Planner Agent. It primarily handles MRI segmentation and dose prediction. Inputs

include task instructions, and imaging data, while outputs are tool-derived results such as lesion masks and dose estimates, which are then passed to the Strategy Agent.

**Segmentation Module** is built on the SAM2 framework, it enables consistent 3D MRI lesion delineation with minimal user input. Three prompt modes: Autonomy, Click, and BBox, to support flexible application scenarios.

**Dose Prediction Module** is a radiomics-based pipeline combines imaging features with clinical variables, using feature selection and XGBoost regression to predict individualized dose values. Model performance is validated through cross-validation metrics.

### 3.3 STRATEGY AGENT

The Strategy Agent is the core reasoning module of FUAS-Agents, responsible for generating personalized treatment plans from multimodal inputs. It integrates segmentation and dose results from the Executor Agent with patient records, MRI reports, and domain knowledge from the knowledge base. The outputs are structured candidate treatment plans that combine imaging outcomes, dose estimates, and FUAS procedural pathways. These plans are then passed to the Optimizer Agent for validation and refinement.

#### 3.3.1 ENSURING DATA QUALITY AND CLINICAL RELEVANCE IN MODEL FINE-TUNING

To ensure data quality and clinical relevance, this study uses real-world patient data, which better reflect clinical practice complexities compared to synthetic datasets Xie et al. (2024). We selected data from collaborating healthcare organizations, following IRB protocols and ensuring privacy protection. The dataset includes treatment reports for over 2,000 patients with hysteromyoma, covering demographics, MRI reports, and ultrasound protocols. The data undergoes cleaning and safety checks with GPT-4, and diagnostic information is extracted for fine-tuning. To ensure validity, five independent experts review and annotate the reports, minimizing biases Zhou et al. (2024). The dataset is split: 10% for validation and 90% for fine-tuning, ensuring relevance and data integrity during training.

#### 3.3.2 FUAS MODEL FINE-TUNING PROCESS

In the fine-tuning process, we utilize Qwen-3 14B as the base model for treatment strategy generation, leveraging its excellent text comprehension and generation capabilities. Additionally, we adopt the LoRA fine-tuning method, which introduces a low-rank adaptation layer to the model, allowing it to specialize in treatment planning while maintaining computational efficiency. The objective of instruction fine-tuning is to minimize the expected loss for each pair $(X, Y)$ within the dataset $\mathcal{D}$, which is formulated in the equation below and optimized by adjusting the parameter set $\theta_{\text{Agent}}$:

$$\theta^*_{\text{Agent}} = \arg\min_{\theta_{\text{Agent}}} \mathbb{E}_{(X,Y)\in\mathcal{D}} \left[ \mathcal{L}(\theta_{\text{Agent}}(X), Y) \right]$$

The key parameters in the fine-tuning process correspond to different input-output mappings, which are described as follows: $\theta_{M2F}$: The input data $X$ consists of the pair $(\mathcal{M}, \mathcal{B})$, where $\mathcal{M}$ represents 3D MRI data with dimensions $3 \times 25 \times 1024 \times 1024$ (3 channels, 25 slices, 1024x1024 images), and $\mathcal{B}$ contains structured patient data, such as demographics and medical history. The target output $Y$ is the feature vector $\mathcal{F}$, which represents the extracted features used for treatment planning; $\theta_{T2F}$: The input $X$ is composed of the pair $(\mathcal{T}, \mathcal{B})$, where $\mathcal{T}$ refers to tool outputs (e.g., segmentation results or dose predictions), and $\mathcal{B}$ is the structured patient data. The target remains the feature vector $\mathcal{F}$, which is used for treatment planning; $\theta_{F2I}$: The input data $X$ is the pair $(\mathcal{F}, \mathcal{G})$, where $\mathcal{F}$ is the feature vector and $\mathcal{G}$ represents clinical guidelines (treatment rules and standards). The target output $Y$ is the interpretable output $\mathcal{I}$, which consists of the final treatment recommendations or reports.

### 3.4 OPTIMIZER AGENT

The Optimizer Agent serves as the quality-control and reflection module of FUAS-Agents, ensuring candidate treatment plans from the Strategy Agent meet clinical standards. It cross-checks dose predictions and strategies against clinical guidelines, principles, and prior cases stored in the Memory

Module. If inconsistencies are detected, the plans are returned for regeneration. Should errors persist beyond $N > 2$ iterations, physician intervention is triggered. This agent works by processing candidate treatment plans as inputs and producing finalized recommendations with revision feedback as outputs.The optimization is modeled as a constrained objective function:

$$\theta_{\text{opt}}^* = \arg \min_\theta \left[ \mathcal{L}_{\text{task}}(\theta) + \lambda \cdot \mathcal{L}_{\text{constraint}}(\theta, \mathcal{G}) \right],$$

Where $\mathcal{L}_{\text{task}}$ measures the alignment between the generated treatment plan and task objectives, such as clinical effectiveness and feasibility. It is computed by evaluating the plan's clinical validity based on the model's internal assessment of expected outcomes. $\mathcal{L}_{\text{constraint}}$ quantifies the deviation from clinical guidelines $\mathcal{G}$ by retrieving relevant guidelines from the Memory Module and comparing them with the treatment plan, ensuring compliance. $\lambda$ is a coefficient that controls the extent to which different guidelines and prior cases are referenced during the RAG process. By adjusting $\lambda$, the model can iteratively adjust the weight given to various sources of information, such as expert guidelines and historical cases, in the treatment plan generation.

Clinical guidelines $\mathcal{G}$ are encoded using the text-embedding-ada-002 model, transforming them into high-dimensional vectors stored in the Memory Module and indexed with FAISS for efficient retrieval. The generated plan input to $\mathcal{L}_{\text{task}}$ includes patient demographics, target regions (MRI), treatment dosage, strategy, sequence, and post-treatment monitoring. An example is provided in Appendix B.

### 3.5 MEMORY MODULE

The Memory Module is the core information support of FUAS-Agents, serving both as a structured knowledge base and a dynamic memory store. It retains authoritative FUAS resources, including clinical guidelines, annotated cases, and expert consensus, while also recording intermediate reflections and optimization feedback for future reuse. Its main function is to provide traceable medical evidence and historical context to the Strategy Agent for plan generation and the Optimizer Agent for compliance checks.The construction pipeline includes: (1) **Curation** of guidelines, consensus, and labeled FUAS cases; (2) **Preprocessing** via OCR, noise filtering, and clinical entity recognition; (3) **Structuring** into semantically tagged JSON entries; (4) **Embedding and indexing** in FAISS for retrieval-augmented generation (RAG).

Queries ($q$) are information requests posed by the Strategy Agent, representing clinical questions that require relevant evidence from the knowledge base. These queries are embedded into vectors $v_i \in \mathbb{R}^d$. Documents ($d$) represent stored knowledge units such as clinical guidelines and expert consensus. After preprocessing, they are embedded into vectors $d_j \in \mathbb{R}^d$ and indexed in FAISS for efficient retrieval.Formally, let the query set be $Q = \{q_1, q_2, \ldots, q_m\}$, with each sub-query embedded as vector $v_i \in \mathbb{R}^d$. These vectors are matched against the document vector set $D = \{d_1, d_2, \ldots, d_n\}$ to retrieve the top-k relevant passages:

$$\text{TopK}(q_i) = \arg \max_{d_j \in D} \cos(v_i, d_j), \quad j = 1, \ldots, n$$

The retrieved results are merged into a unified evidence set:

$$E = \text{Merge}(\text{TopK}(q_1), \text{TopK}(q_2), \ldots, \text{TopK}(q_m))$$

Finally, the Strategy Agent synthesizes a reasoning path $S$ from the evidence $E$, which is passed to the Optimizer Agent for quality verification.

## 4 EXPERIMENT

### 4.1 OVERVIEW

This section evaluates the FUAS-Agents framework. Section 4.1 covers the MRI-based segmentation model, Section 4.2 discusses dose prediction using radiomics, and Section 4.3 details treatment

strategy generation with a multimodal large language model. It also presents expert evaluations of the treatment plans for clinical validation, followed by ablation experiments to confirm system effectiveness. An example of the treatment plan is provided in Appendix B.

## 4.2 SEGMENTATION MODULE

### 4.2.1 MODELS

In this study, we use Medical SAM 2 (MedSAM-2) as the foundational model, a robust segmentation framework that extends SAM 2 for medical image segmentation in 3D medical images. We establishe baseline performance by evaluating MedSAM-2 on the validation dataset across three prompt types : Autonomy, Click and BBox.

Segmentation experiments are performed on 3D MRI scans and the corresponding manual segmentations of 702 patients undergoing FUAS treatment. The dataset is partitioned into training dataset (n = 561) and validation dataset (n = 141) using an 8:2 split, ensuring comparable segmentation difficulty distributions between sets.

Table 1: Quantitative evaluation of MedSAM-2 and our method across different prompt types. Our method consistently outperforms MedSAM-2 across all prompt types, with the most significant improvement observed for the "Autonomy" and "Click" prompt types. The bolded numbers indicate the best performance for each metric.

| Prompt Type | Model | Dice | IoU | Average |
|---|---|---|---|---|
| Autonomy | MedSAM-2 | 0.1577 | 0.1170 | 0.1374 |
| Autonomy | Ours | 0.6645 | 0.6245 | 0.6445 |
| Click | MedSAM-2 | 0.7670 | 0.7245 | 0.7458 |
| Click | Ours | **0.8550** | **0.8085** | **0.8318** |
| BBox | MedSAM-2 | 0.7596 | 0.7391 | 0.7494 |
| BBox | Ours | 0.7724 | 0.7574 | 0.7649 |

### 4.2.2 RESULT

The segmentation performance of our model, measured by Dice score and IoU, under different prompt types is shown in Table 1. For the "Autonomy" prompt, our model improves segmentation by over 4.5× compared to the baseline, demonstrating strong zero-input generalization. The "Click" prompt yields an 11.5% improvement over MedSAM-2, while the "BBox" prompt also shows modest gains across all metrics. These results highlight our method's superiority over MedSAM-2, particularly in zero-input segmentation scenarios.The detailed segmentation results can be found in Appendix C.

## 4.3 DOSE PREDICTION

### 4.3.1 BASELINES

MRI data from 149 FUAS-treated patients (Cases with poor image quality, combined treatment with auxiliary drugs such as anhydrous ethanol, postoperative non-perfusion volume ratio less than 70%, and other uterine or adnexal lesions were excluded, n=93) were analyzed (mean age 41.2 ± 11.3). Each case included T2WI sequences and corresponding dose records (15–245 kJ). The dataset was split 8:2 into training ($n = 120$) and validation ($n = 29$) sets, with no significant intergroup dose distribution difference ($P = 0.74$, K-S test).

### 4.3.2 FEATURE EXTRACTION AND RADIOMICS ANALYSIS

Standardized 3D lesion annotations were performed using ITK-SNAP, followed by extraction of 107 radiomic features via PyRadiomics, including first-order statistics, shape, and texture descriptors (GLCM, GLSZM, GLRLM). Lasso regression identified 10 key predictors (Shapley range: 0.08–0.21), with the top three being: (1) GLCM_ClusterShade (21.3%), (2) GLRLM_LongRunEmphasis

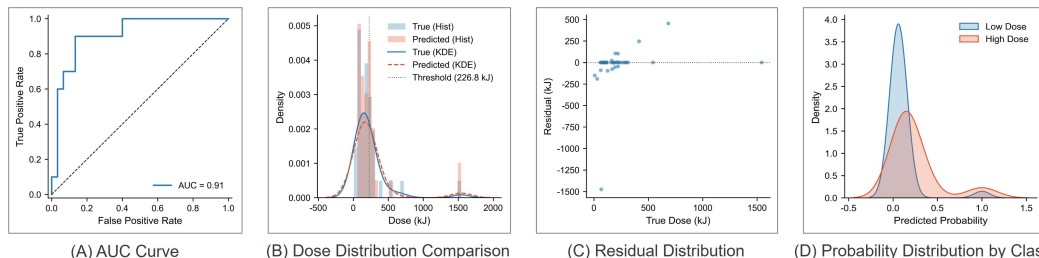

(A) AUC Curve     (B) Dose Distribution Comparison     (C) Residual Distribution     (D) Probability Distribution by Class

Figure 3: **Comprehensive result of model performance in dose prediction.** (A) AUC Curve Analysis. (B) Dose Distribution Analysis. (C) Residual Analysis. (D) The bimodal probability density curve.

(18.7%), and (3) GLCM_ShortRunHighGrayLevelEmphasis (15.2%). Texture complexity showed a positive correlation with dose ($r = 0.62$, $P < 0.01$).

### 4.3.3 MODEL RESULT

The dose prediction model performed well (AUC = 0.91), with ROC analysis confirming high sensitivity and specificity. Ablation and baseline comparisons validate the design choice of the proposed module. Predicted dose distributions closely matched ground truth (KS test $P = 0.76$), though the model was slightly conservative in high-dose regions ($\chi^2 = 5.32$, $P = 0.021$), showing a minor peak around 1500 kJ and underestimation above 300 kJ. Residuals were typically within ±50 kJ at lower doses. Clinicians confirmed the current MAE is acceptable. Risk classification showed a clear bimodal pattern (KL divergence = 0.83), supporting effective stratification.

Table 2: Model Performance Comparison

| Model | ROUGE | | | BLEU | | | |
|---|---|---|---|---|---|---|---|
| | **R-1** | **R-2** | **R-L** | **B-1** | **B-2** | **B-3** | **B-4** |
| GPT-4 | 0.3291 | 0.1339 | 0.2359 | 0.3572 | 0.1192 | 0.0551 | 0.0342 |
| ChatGPT-4o | 0.3535 | 0.1121 | 0.1713 | 0.2004 | 0.0660 | 0.0288 | 0.0163 |
| Claude 3 Sonnet | 0.3483 | 0.1004 | 0.1864 | 0.2198 | 0.0628 | 0.0258 | 0.0155 |
| LLaMA 4 Scout 17B | 0.3480 | 0.1159 | 0.1952 | 0.1952 | 0.2717 | 0.0830 | 0.0386 |
| DeepSeek-V3 | 0.3655 | 0.1137 | 0.1927 | 0.2114 | 0.0651 | 0.0291 | 0.0172 |
| DeepSeek-R1 | 0.3368 | 0.0942 | 0.1581 | 0.1672 | 0.0470 | 0.0181 | 0.0094 |
| GLM-4-32B | 0.3234 | 0.0934 | 0.1374 | 0.1219 | 0.0412 | 0.0169 | 0.0087 |
| Moonlight 16B | 0.3419 | 0.1174 | 0.1999 | 0.2431 | 0.0760 | 0.0343 | 0.0208 |
| Yi-34B-Chat | 0.3570 | 0.1130 | 0.1884 | 0.2265 | 0.0697 | 0.0298 | 0.0187 |
| Qwen3-14B | 0.3329 | 0.0823 | 0.1291 | 0.1191 | 0.0329 | 0.0129 | 0.0073 |
| Doubao-1.5-pro | 0.3102 | 0.0725 | 0.1298 | 0.1314 | 0.0344 | 0.0135 | 0.0076 |
| FUAS | **0.5512** | **0.3267** | **0.4269** | **0.4988** | **0.2765** | **0.1806** | **0.1300** |

### 4.4 REPORT GENERATION

### 4.4.1 BASELINES

We select multiple baseline models to evaluate the performance of our proposed FUAS model, categorized into closed-source and open-source. The closed-source models include ChatGPT-4o-Latest and Claude-3-7-Sonnet-Latest, both of which excel in natural language understanding and generation. The open-source models consist of GLM-4-32B, Doubao-1.5-thinking-pro, Qwen3-14B, DeepSeek-R1, DeepSeek-V3, Yi-34B-Chat, Llama-4-Scout-17B-16E-Instruct, and Moonlight-16B-A3B-Instruct, which demonstrate strong performance in medical tasks. To ensure fairness, the se-

Table 3: Ablation Study Results

| Group | Completeness | Accuracy | Fluency | Compliance |
|---|---|---|---|---|
| Full-Function Group (Baseline) | 82.5% | 80.0% | 87.5% | 97.5% |
| Ablation Group 1 (No Executor) | 36.3% | 77.5% | 82.5% | 92.5% |
| Ablation Group 2 (No Optimizer) | 82.5% | 65.0% | 77.5% | 72.5% |
| Ablation Group 3 (No Memory) | 76.3% | 58.8% | 73.8% | 65.0% |

lected baseline models are chosen to have parameters and functionalities similar to or higher than those of our FUAS base model, allowing us to fully showcase the potential of each model.

### 4.4.2 TRAINING DETAILS

For fine-tuning, we use the LoRA method with a rank of 8 and a scaling factor of 16. The learning rate is set to $5 \times 10^{-5}$, with training conducted over 3 epochs. A maximum gradient norm of 1.0 ensures training stability. The batch size per GPU is 2, and the learning rate is adjusted with a cosine scheduler. Training is done with BF16 precision to optimize memory usage, and a 4-step warm-up period gradually increases the learning rate. The training is distributed across four NVIDIA A800 80GB GPUs for faster processing.

### 4.4.3 MODEL COMPARISON AND EVALUATION METRICS

To evaluate the performance of the **FUAS model**, we test it using 200 randomly selected samples (10% of the dataset). All models are assessed under identical conditions and tasked with generating structured treatment strategies. Their outputs are evaluated using standard metrics: **ROUGE-1**, **ROUGE-2**, **ROUGE-L**, and **BLEU-1** to **BLEU-4**, which measure recall and n-gram precision in text generation. The results show that the **FUAS model** outperforms all baseline models, achieving the highest ROUGE and BLEU scores. Compared to models like **GPT-4** and **GLM-4-32B**, **FUAS's performance is significantly better**, demonstrating its ability to match or exceed larger models even with fewer parameters. This highlights the **FUAS model's** effectiveness and efficiency for treatment strategy generation, making it a valuable tool for real-world medical applications. The comparative results are shown in Table 2.

### 4.4.4 CASE STUDY

In this section, we assess the performance of several large models in generating treatment plans. As shown in Appendix E, ChatGPT, GLM-4, DeepSeek-R1, and GPT-4O provide treatment frameworks, but differ in refinement and personalization. ChatGPT and GLM-4 focus on standard processes without adjustments, while DeepSeek-R1 offers preoperative preparation but overlooks complex conditions. In contrast, FUAS excels in diagnosis accuracy, treatment adjustments, and risk control, providing personalized plans that align more closely with patient needs in preoperative and postoperative evaluations.

### 4.5 ABLATION STUDIES

We conducted an ablation study to evaluate the contribution of each module in FUAS-Agents. The Full-Function Group (Baseline) includes all five parts: Planner, Executor, Strategy, Optimizer, and Memory. In the ablation study, we removed one agent at a time: Ablation Group 1 removes the Executor Agent, Ablation Group 2 removes the Optimizer Agent, and Ablation Group 3 removes the Memory Module. Twenty test cases with over 90% completeness were selected, and four senior FUAS experts evaluated the generated treatment plans based on completeness, accuracy, fluency, and compliance. The results are shown in Table 3.

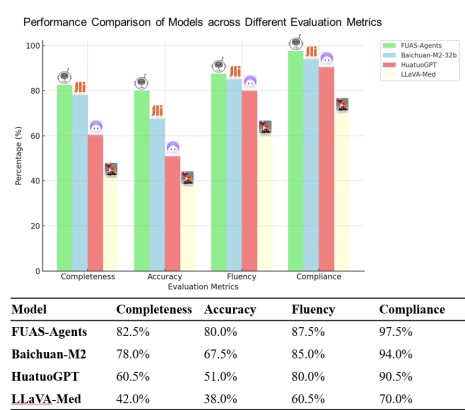

Figure 4: **Human evaluation across different models.**

| Model | Completeness | Accuracy | Fluency | Compliance |
|-------|-------------|----------|---------|------------|
| **FUAS-Agents** | 82.5% | 80.0% | 87.5% | 97.5% |
| **Baichuan-M2** | 78.0% | 67.5% | 85.0% | 94.0% |
| **HuatuoGPT** | 60.5% | 51.0% | 80.0% | 90.5% |
| **LLaVA-Med** | 42.0% | 38.0% | 60.5% | 70.0% |

## 4.6 EFFICIENCY ANALYSIS

We evaluate the efficiency of FUAS-Agents by examining runtime, token usage, and success rate across the Planner, Executor, Strategy, and Optimizer Agents. Runtime measures task completion time, token usage reflects computational cost, and success rate indicates expert-validated task quality. Detailed results are reported in Appendix F.

## 4.7 HUMAN EVALUATION

To complement the quantitative analysis, four senior FUAS specialists independently evaluated 20 randomly selected cases based on standardized criteria. The assessments covered four dimensions: completeness, accuracy, fluency, and compliance. Results, presented in Appendix G, show that FUAS-Agents performed well across all metrics. Specifically, 82.5% of the plans were comprehensive in completeness, over 80% were clinically appropriate for accuracy, 87.5% were coherent for fluency, and 97.5% adhered to ethical, safety, and regulatory standards for compliance.

To further validate the robustness and clinical applicability of the proposed system, we also conducted a comparative evaluation with three representative large-scale medical language models (Baichuan-M2, Huatuo GPT, and LLaVA-Med). Each model was required to generate a complete FUAS treatment planning. Subsequently, human experts scored the models. As shown in Figure 4, the comparison results indicate that FUAS-Agents consistently outperformed the other models, highlighting its advantage in generating FUAS treatment planning.

## 5 CONCLUSION

In this work, we present FUAS-Agents, a modular multi-agent system powered by multimodal LLMs for generating personalized treatment planning in Focused Ultrasound Ablation Surgery. The system integrates image segmentation, dose prediction, and therapeutic strategy formulation into a unified workflow that mirrors the physician's reasoning process. Experimental results demonstrate that FUAS-Agents achieves superior performance compared with open-source baselines and has been positively evaluated by senior clinicians.

Despite these promising results, this study remains limited by its reliance on single-center data. Future work will extend validation to multi-center cohorts to improve generalizability and robustness. In addition, the deployment of agent systems in clinical practice involves ethical and operational challenges. FUAS-Agents adopts a human-in-the-loop framework to ensure physician oversight under uncertainty. Future work will incorporate self-monitoring and out-of-distribution detection to enhance reliability and interpretability. Patient privacy remains a priority, with federated learning envisaged to secure multi-center data. Fairness and accountability will be addressed through data diversification and bias auditing.

**Ethics Statement.** This study uses retrospective, anonymized single-center MRI data with institutional approval, ensuring that no personally identifiable information is disclosed. FUAS-Agents is designed under a human-in-the-loop paradigm, where physicians maintain oversight of automated recommendations. Patient privacy, fairness, and accountability are prioritized, with future work exploring federated learning and bias auditing to further safeguard ethical compliance.

**Reproducibility Statement.** We provide detailed descriptions of model design, training, and evaluation in the main text and appendix. Source code is publicly available at the anonymous repository (see abstract). Due to privacy restrictions, raw clinical data cannot be shared, but synthetic examples and interfaces are provided to support reproducibility.

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

# A    EXPLANATIONS OF FORMULA SYMBOLS

Table A1: Notation Definitions

| Notation | Description |
|---|---|
| M2F, T2F, F2I | Transformation of different data types through the fine-tuning process. |
| M2F | Maps MRI data (M) and patient metadata to features (F) for treatment planning. |
| T2F | Maps tool outputs (e.g., segmentation results, dose predictions) to features (F). |
| F2I | Transforms feature representations (F) into interpretable outputs (I), such as final treatment recommendations. |

Table A2: Data Types and Structures

| Data Type | Input Description | Output Description |
|---|---|---|
| Medical Image Data (M) | 3D MRI image data with dimensions $(3 \times 25 \times 1024 \times 1024)$ | Processed image data generating segmentation masks or feature representations (F) for treatment planning. |
| Dose Prediction Data | Input: predicted dose (float), weight (joblib), MRI image (nii), and mask (nii). | Output: Predicted dose values and associated weights. |
| Clinical Data (B) | Structured patient data with $\sim$50 features (demographics, medical history). | Integrated data for the Strategy Agent. |
| Feature Vector (F) | High-dimensional features (512 or 1024 dims) extracted from M or T. | Input for dose prediction or treatment planning. |
| Treatment Plan (T) | Structured parameters (10 key parameters: dose, area, etc.). | Final treatment plan. |
| Clinical Guidelines (G) | Expert-defined treatment standards and rules. | Ensures optimized, compliant treatment plans. |
| Interpretative Output (I) | Model-generated treatment plans. | Final human-readable recommendations or reports. |

# B    ILLUSTRATIVE EXAMPLE OF FUAS-AGENTS TREATMENT PLANNING

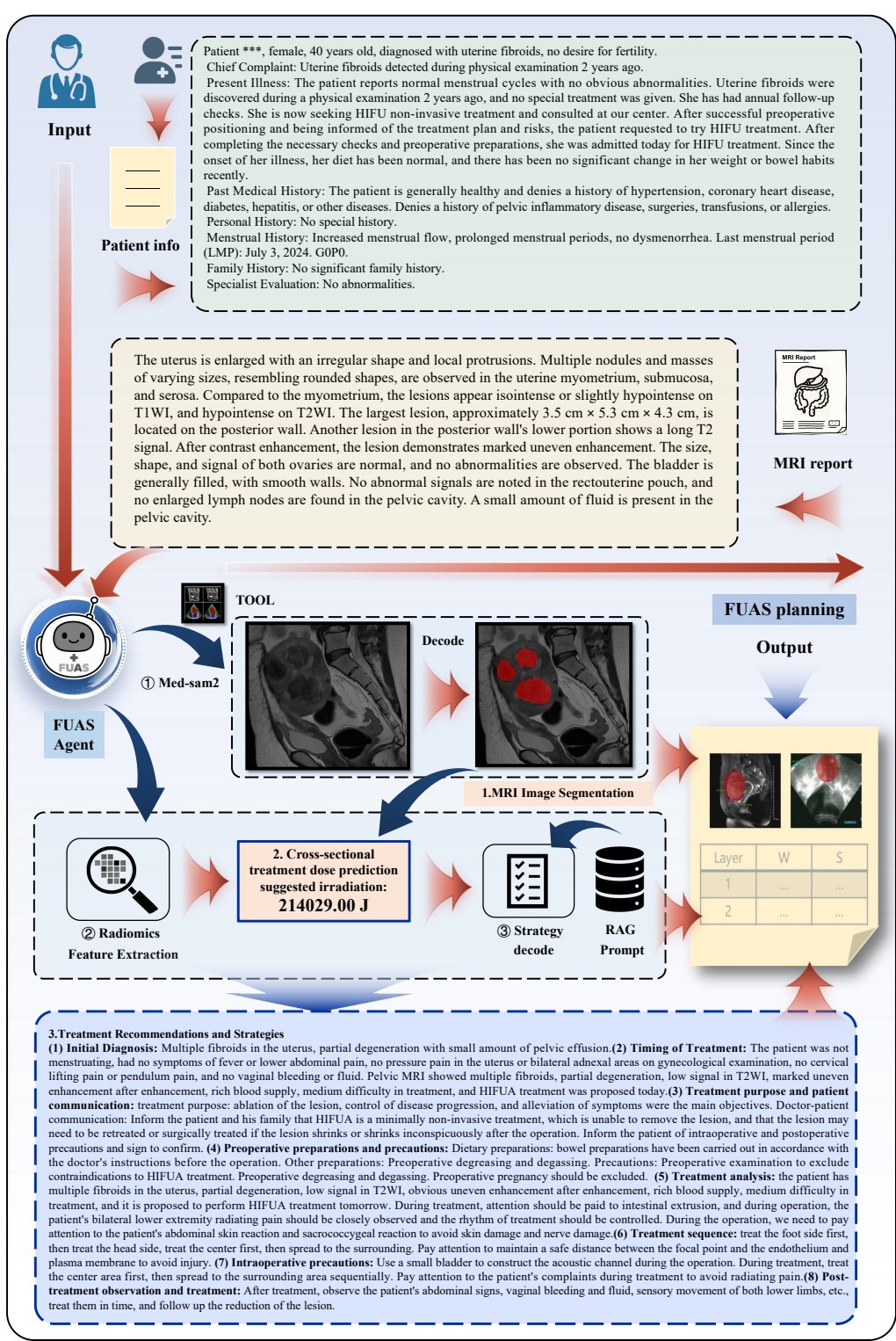

Figure 5: Illustrative example of FUAS-Agents treatment planning

## C  SAMPLE

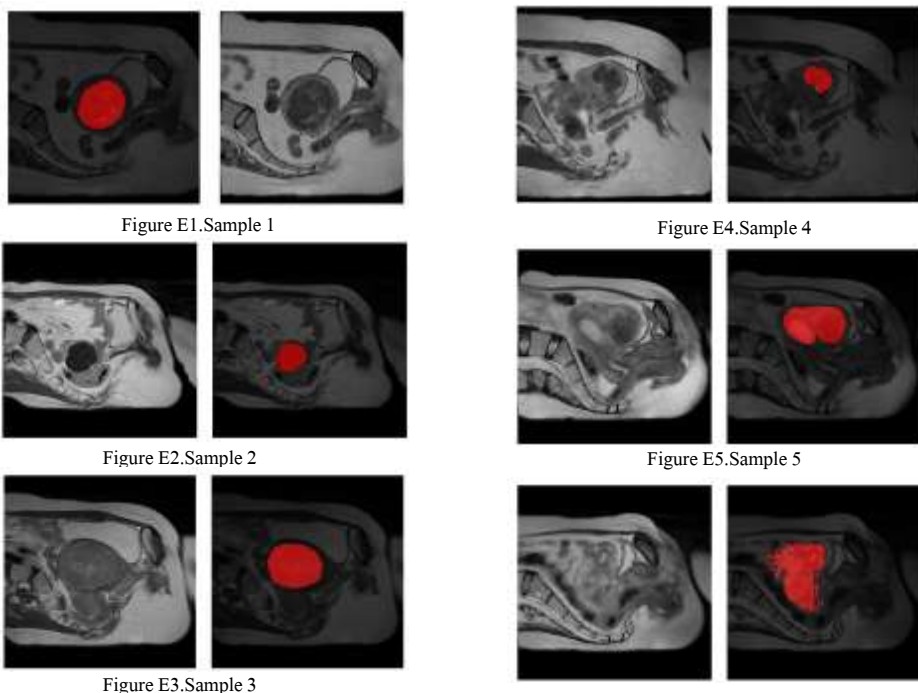

Figure E1.Sample 1

Figure E4.Sample 4

Figure E2.Sample 2

Figure E5.Sample 5

Figure E3.Sample 3

Figure E6.Sample 6

Figure C1: Comprehensive visualization for Appendix E

## D  SUPPLEMENTARY INFORMATION ON DOSE PREDICTION

1. Supplementary Dose Prediction Experiments

| Model | MAE(kJ) | AUC | R² |
|---|---|---|---|
| Lasso Grid Geometric Features (baseline module) | 128.9 | 0.67 | 0.65 |
| XGBoost Bayes FinFeat Radiomics | 28.4 | 0.86 | 0.75 |
| Support Vector Regressor Radiomics | 37.9 | 0.82 | 0.70 |
| Random Forest Radiomics | 42.3 | 0.83 | 0.72 |
| Lasso-XGBoost Radiomics Dose Predictor | 27.0 | 0.91 | 0.82 |

2. Technical Details of the Dose Prediction Process

(1) Data preparation and feature extraction Experienced clinicians manually delineated the target lesions on NIfTI-format medical images. Quantitative radiomic features were extracted from both the images and segmentation masks using the pyradiomics toolkit.

(2) Feature preprocessing and selection All features underwent variance filtering and Z-score normalization. To identify stable predictors, Lasso regression was repeatedly applied on random data subsets, and features were ranked by their selection frequency. Based on this stability selection process, a LassoCV model was further used to shrink coefficients and obtain the final feature set.

(3) Model construction A unified sklearn.pipeline.Pipeline connected feature selection, normalization, and regression. The final predictor adopted an XGBoost regressor, whose hyperparameters were optimized via grid-search cross-validation.

(4) Hyperparameter settings StabilitySelector threshold: [0.5, 0.6]; LassoCV alphas: logspace(3, 1, 15); XGBoost:
$\max_d epth = [3, 5], learning_r ate = 0.01, reg_a lpha = 0, reg_l ambda = 1, gamma = 0.$

(5) Definition of high-dose vs. low-dose groups The quartile cut-points of the training-set dose distribution were used to define dose categories.

(6) Software and computational environment The workflow was implemented using pyradiomics, XGBRegressor, SimpleITK, LassoCV, GridSearchCV, and StabilitySelector.

# E CASE STUDY OF TREATMENT PLANNING GENERATION BY DIFFERENT LLMS

**Prompt：** Please give a treatment plan for the individual corresponding to the MRI report based on the principle content of ultrasound surgery for uterine fibroids.The content of the treatment plan should include:1.Initial diagnosis:2.Timing of treatment: 3.Purpose of treatment and doctor-patient communication:4.Pre-operative preparations and precautions:5.Analysis of the treatment and intra-operative precautions:6.Post-therapeutic observation and management.

**Input：** The patient's MRI report is as follows: posterior uterus, enlarged in size, size about 64*81*60mm, uterine wall can be seen multiple abnormal signal foci, the largest is located in the left side wall, size about 56*46*50mm, T1 isosignal, T2 low signal, and enhancement is inhomogeneous and obvious enhancement.

**1. Preliminary diagnosis: multiple smooth leiomyomas of the uterus.**
**2. Timing of treatment:** the patient is not in menstrual period, at present, there is no obvious fever, lower abdominal pain symptoms, no vaginal bleeding, gynecological examination of the uterus, bilateral adnexal pressure, and the rest of the relevant examination did not see obvious abnormalities, and it is proposed to carry out the HIFUA treatment under sedation and analgesia tomorrow.
**3. Therapeutic purpose and patient-doctor communication:** the patient's uterine multiple smooth leiomyomas, and the current treatment is aimed at the ablation of leiomyoma foci and control of leiomyoma growth. 4. The purpose of this treatment is to ablate the leiomyoma foci and control the growth of leiomyoma. Inform the patient and her family that FUAS is a non-invasive treatment for fibroid lesions that are visible under the patient's current ultrasound and the acoustic channel is safe, and that intraoperative pathology examination is not possible, and that there is a possibility that the fibroid shrinks or doesn't continue to grow after the operation, and there is a possibility of recurrence after the operation, and that if there is a recurrence, the patient can consider FUAS or surgery again; the above has already been communicated with the patient and her family, who have expressed their understanding and asked for the treatment of HIFUA.
**4. Preoperative preparation and precautions:** at present, strict bowel preparation has been carried out, preoperative degreasing and degassing.
**5. Treatment analysis and intraoperative precautions:** the patient's posterior uterus, multiple leiomyomas, it is proposed to construct the acoustic channel with the middle bladder + middle water bladder. Treatment sequence: posterior wall leiomyoma - anterior wall leiomyoma - left wall leiomyoma - right wall leiomyoma. Pay attention to the safety of cephalad acoustic channel during treatment, and observe the patient's double lower limbs reaction to avoid nerve injury.
**6. Post-treatment observation and treatment:** observe the patient's abdominal signs, abdominal skin, and double lower limbs activity after operation. Follow-up on the lesion shrinking or not continuing to grow.

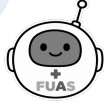

**FUAS**

Figure E1: Treatment Plan Generated by FUAS-Agents

**Prompt**：Please give a treatment plan for the individual corresponding to the MRI report based on the principle content of ultrasound surgery for uterine fibroids.The content of the treatment plan should include:1.Initial diagnosis:2.Timing of treatment: 3.Purpose of treatment and doctor-patient communication:4.Pre-operative preparations and precautions:5.Analysis of the treatment and intra-operative precautions:6.Post-therapeutic observation and management.

**Input**：The patient's MRI report is as follows: posterior uterus, enlarged in size, size about 64*81*60mm, uterine wall can be seen multiple abnormal signal foci, the largest is located in the left side wall, size about 56*46*50mm, T1 isosignal, T2 low signal, and enhancement is inhomogeneous and obvious enhancement.

**1. Initial Diagnosis:**Uterine fibroid (left side wall, 6.57.06.1cm), ovarian cyst, cervical nabothian cyst.
**2. Timing of Treatment:**Considering the size and location of the fibroid and the absence of obvious symptoms, the treatment can be scheduled 1-2 weeks after menstruation for HIFU treatment.
**3. Treatment Objective and Doctor-Patient Communication:**The goal of the treatment is to use HIFU to ablate the uterine fibroid, relieve symptoms, and improve quality of life. During doctor-patient communication, it is necessary to inform the patient about the principles of HIFU treatment, the process, possible risks and complications, and the potential reactions after the treatment along with the follow-up plan.
**4. Preoperative Preparation and Precautions:**(1)Check the patient's history for abortion within the last 3 months, IUD, pelvic inflammatory disease, hearing/communication disorders, etc.(2)If the patient has an IUD, it should be removed 3 days before the HIFU treatment.(3)Check for any history of lower abdominal surgery to determine if there is severe intestinal adhesion or scarring.(4)The patient should undergo relevant tests, such as routine blood tests, urinalysis, and coagulation function.(5)The patient needs to sign an informed consent form.
**5. Treatment Analysis and Intraoperative Precautions:**(1)According to the MRI report, the uterine fibroid is located on the left side wall, showing T2 high signal and significant enhancement, indicating rich blood supply, making the treatment more difficult.(2)During treatment, attention should be paid to the distance between the focus and the endometrium, ensuring it is greater than 15mm.(3)Patient response should be monitored during the procedure, and the dosage and intensity should be adjusted according to the gray-scale changes in the treatment target area.(4)Considering the patient's anteverted uterus, a small bladder can be used during treatment.(5)Protect the intestines and bladder during treatment.
**6. Post-Treatment Observation and Management:**
(1)After treatment, the patient should rest in the observation room for 30 minutes, with monitoring of vital signs.(2)Follow-up should be performed 1-2 weeks post-treatment to check symptoms and signs.
(3)An MRI should be done 3-6 months after the treatment to assess the treatment effect.(4)Adjust the follow-up treatment plan based on the patient's condition.Please note that the above treatment plan is for reference only. The specific treatment plan should be adjusted based on the patient's actual condition and the doctor's experience.

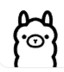

**LLaMA 4 Scout 17B**

Figure E2: Treatment Plan Generated by LLaMA

**Prompt：** Please give a treatment plan for the individual corresponding to the MRI report based on the principle content of ultrasound surgery for uterine fibroids.The content of the treatment plan should include:1.Initial diagnosis:2.Timing of treatment: 3.Purpose of treatment and doctor-patient communication:4.Pre-operative preparations and precautions:5.Analysis of the treatment and intra-operative precautions:6.Post-therapeutic observation and management.

**Input：** The patient's MRI report is as follows: posterior uterus, enlarged in size, size about 64*81*60mm, uterine wall can be seen multiple abnormal signal foci, the largest is located in the left side wall, size about 56*46*50mm, T1 isosignal, T2 low signal, and enhancement is inhomogeneous and obvious enhancement.

Here's the translation structured according to your request:
**1. Preliminary Diagnosis:**Based on the MRI report, the patient is diagnosed with a subserosal fibroid located on the left posterior  wall of the uterus.
**2. Treatment Timing:**Considering the size and location of the fibroid, this is the appropriate time to proceed with treatment.
**3. Treatment Goals and Doctor-Patient Communication:**The main goal of treatment is to eliminate the fibroid, alleviate symptoms, and improve the patient's quality of life through HIFU (High-Intensity Focused Ultrasound) surgery. It is important to have thorough communication with the patient, explaining the necessity of the surgery, potential risks, and expected outcomes.
**4. Preoperative Preparation and Precautions:**A comprehensive health evaluation, including blood tests and an ECG, should be conducted preoperatively. The patient should fast for 8 hours prior to the surgery, starting from the night before. Adequate bladder filling should be ensured by instructing the patient to empty their bladder before the procedure.
**5. Treatment Analysis and Intraoperative Considerations:**Since the fibroid is located on the left posterior wall subserosally, the treatment may be challenging. Care should be taken during the procedure to protect surrounding normal tissues and avoid damage. The fibroid should be treated from the center outward. The dose and intensity should be adjusted based on the patient's response and the changes in the target area's gray scale.
**6. Post-Treatment Observation and Management:**After the procedure, close monitoring of the patient's vital signs is essential, with attention to potential bleeding, pain, or other complications. Based on the postoperative condition, medications such as painkillers and antibiotics should be administered. The patient will need regular follow-ups to assess the treatment's effectiveness.

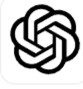

**GPT-4**

Figure E3: Treatment Plan Generated by GPT-4

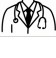

**Prompt：** Please give a treatment plan for the individual corresponding to the MRI report based on the principle content of ultrasound surgery for uterine fibroids.The content of the treatment plan should include:1.Initial diagnosis:2.Timing of treatment: 3.Purpose of treatment and doctor-patient communication:4.Pre-operative preparations and precautions:5.Analysis of the treatment and intra-operative precautions:6.Post-therapeutic observation and management.
**Input：** The patient's MRI report is as follows: posterior uterus, enlarged in size, size about 64*81*60mm, uterine wall can be seen multiple abnormal signal foci, the largest is located in the left side wall, size about 56*46*50mm, T1 isosignal, T2 low signal, and enhancement is inhomogeneous and obvious enhancement.

**1. Preliminary Diagnosis:**The patient has been diagnosed with a submucosal fibroid located on the right side of the uterus, measuring approximately 34×33×30mm. The MRI shows a T1-equivalent signal, low T2 signal, and significant enhancement after contrast. The uterus is in a retroverted position. These characteristics indicate a fibroid with considerable blood supply, making it suitable for HIFU treatment.
**2. Treatment Timing:**It is recommended to perform the treatment outside of menstruation and ovulation periods, ideally within 3-7 days after the end of menstruation. If the patient has a history of abortion within the last three months, pelvic inflammation, an IUD in place, hearing or communication disabilities, or a history of major abdominal surgeries, HIFU treatment should be postponed or avoided. Pre-treatment confirmation should ensure there is no active pelvic inflammation, IUD, or severe intestinal adhesions, ensuring the safety of the treatment.
**3. Treatment Purpose and Doctor-Patient Communication:**The main goal of HIFU treatment is to reduce the fibroid's size, alleviate symptoms such as heavy menstrual bleeding, dysmenorrhea, and compression symptoms, and preserve the uterus to avoid surgical trauma. When communicating with the patient, it should be explained that HIFU is a non-invasive treatment, which may require multiple sessions, especially for fibroids with abundant blood supply or deep locations. Special attention should be paid to the distance from the endometrium when treating submucosal fibroids to avoid endometrial damage. Post-treatment, mild abdominal pain, fever, and vaginal discharge may occur, which are normal, and the treatment's effect should be evaluated with a follow-up MRI.
**4. Preoperative Preparation and Precautions:**Preoperative preparation should include ensuring a full bladder to help improve the acoustic pathway, especially since the uterus is retroverted. A large water bag may be used to push the intestines to establish a safe acoustic pathway. Preoperative checks should exclude pregnancy, evaluate pelvic inflammation, and assess intestinal position and adhesions. Fasting for 6 hours and no water intake for 4 hours before surgery is required, and enema or laxatives should be used to reduce intestinal interference. Preoperative informed consent must be obtained.
**5. Treatment Analysis and Intraoperative Considerations:**This patient has a submucosal fibroid with relatively rich blood supply and moderate treatment difficulty. The distance from the endometrium should be at least 15mm to ensure safety; if not, careful consideration should be made as to whether treatment is appropriate. Treatment should prioritize the fibroid's lateral, front-back, and lower areas. The treatment will use a "heart-extraction" irradiation method, advancing layer by layer, observing gray-scale changes, and adjusting the dose based on the patient's response. Intraoperatively, care must be taken to avoid focusing too close to the endometrium to prevent damage, and the intestinal position should be monitored to avoid the acoustic beam passing through the intestines.
**6. Postoperative Observation and Management:**Postoperatively, vital signs, abdominal pain, fever, and vaginal discharge should be monitored, and necessary pain relief and anti-inflammatory treatment should be provided. The patient should avoid food for the first 6 hours and gradually resume a diet. Rest should be maintained for 24 hours, avoiding strenuous activity. An MRI follow-up is scheduled for 1 month to assess the treatment effect. If the fibroid size reduction is insufficient or symptoms persist, further treatment may be considered. Regular follow-ups will be necessary to assess fibroid changes and clinical symptom improvement.

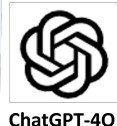

ChatGPT-4O

Figure E4: Treatment Plan Generated by ChatGPT-4o

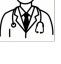

**Prompt：** Please give a treatment plan for the individual corresponding to the MRI report based on the principle content of ultrasound surgery for uterine fibroids.The content of the treatment plan should include:1.Initial diagnosis:2.Timing of treatment: 3.Purpose of treatment and doctor-patient communication:4.Pre-operative preparations and precautions:5.Analysis of the treatment and intra-operative precautions:6.Post-therapeutic observation and management.
**Input：** The patient's MRI report is as follows: posterior uterus, enlarged in size, size about 64*81*60mm, uterine wall can be seen multiple abnormal signal foci, the largest is located in the left side wall, size about 56*46*50mm, T1 isosignal, T2 low signal, and enhancement is inhomogeneous and obvious enhancement.

**1. Preliminary Diagnosis:**Right Wall Submucosal Fibroid (approximately 34×33×30mm), T1 signal equivalent, T2 low signal, significant enhancement after contrast, suggesting rich blood supply.
**2. Treatment Timing:**Elective HIFU Treatment. The following contraindications must be excluded:No history of induced abortion within the past 3 months.No pelvic inflammatory disease or severe abdominal intestinal adhesions/scarring (if the patient has a history of abdominal surgery, ensure the scar width is <15mm).Intrauterine device (IUD) should be removed 3 days before surgery (if present).
**3. Treatment Purpose and Doctor-Patient Communication:**Treatment Purpose: Reduce fibroid size, alleviate symptoms such as excessive menstrual bleeding, and reduce the risk of subsequent surgeries.Key Points for Doctor-Patient Communication:Submucosal fibroids have a special location and require strict protection of the endometrium (focus distance from the endometrium ≥15mm).The rich blood supply may affect ablation efficiency, and multiple treatments or combined postoperative medication (e.g., GnRH-a) may be required.
Risks of complications: skin burns, lower abdominal pain, vaginal spotting, etc.
**4. Preoperative Preparation and Precautions:**Preoperative Preparation:Intestinal Preparation: Low-residue diet 3 days before surgery, morning enema on the day of surgery.Skin Preparation: Degreasing and depilation of the lower abdomen.Bladder Training: Confirm the patient's ability to cooperate with bladder management (retroverted uterus requires moderate bladder fullness to displace the intestines).Precautions:Fasting for 6 hours before treatment, leaving a urinary catheter in place to control bladder volume.Due to the complex acoustic pathway in a retroverted uterus, a larger tension water bag is needed to adjust the pathway and avoid intestinal interference.
**5. Treatment Analysis and Intraoperative Considerations:**Treatment Strategy:Layered Ablation: First, ablate the center of the fibroid (1/4 of the lower part along the vertical axis) using a "core extraction" strategy and gradually extend outward.Dose Adjustment: Due to the rich blood supply (significant enhancement), start with a lower initial dose (e.g., 60-80W) and gradually increase energy after gray-scale changes (formation of lumps).
Safety Boundaries: Ensure the ablation zone is at least 15mm away from the endometrium and at least 10mm away from the serosal surface.Intraoperative Considerations:Monitor gray-scale changes closely to avoid excessive treatment that could damage the endometrium.Adjust the water bag and bladder volume dynamically in the retroverted uterus to ensure the acoustic pathway avoids the intestines.If the patient's pain score exceeds 4, suspend treatment, adjust the focus position, or lower energy.
**6. Postoperative Observation and Management:**Key Postoperative Observations:Monitor for abdominal pain, vaginal bleeding, fever, etc., for the first 24 hours after surgery.Follow-up MRI at 1 month to assess the ablation rate (goal: >50% volume reduction).Management:If menstrual bleeding remains excessive after surgery, consider combined oral contraceptives or GnRH-a therapy for 3 months.If ablation is insufficient, consider a second HIFU treatment or hysteroscopic surgery 3 months later.

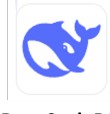

**DeepSeek-R1**

Figure E5: Treatment Plan Generated by DeepSeek

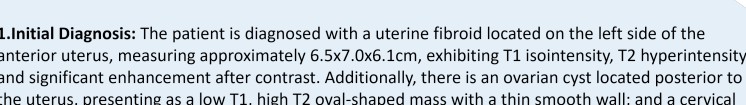

**Prompt：** Please give a treatment plan for the individual corresponding to the MRI report based on the principle content of ultrasound surgery for uterine fibroids.The content of the treatment plan should include:1.Initial diagnosis:2.Timing of treatment: 3.Purpose of treatment and doctor-patient communication:4.Pre-operative preparations and precautions:5.Analysis of the treatment and intra-operative precautions:6.Post-therapeutic observation and management.

**Input：** The patient's MRI report is as follows: posterior uterus, enlarged in size, size about 64*81*60mm, uterine wall can be seen multiple abnormal signal foci, the largest is located in the left side wall, size about 56*46*50mm, T1 isosignal, T2 low signal, and enhancement is inhomogeneous and obvious enhancement.

**1.Initial Diagnosis:** The patient is diagnosed with a uterine fibroid located on the left side of the anterior uterus, measuring approximately 6.5x7.0x6.1cm, exhibiting T1 isointensity, T2 hyperintensity, and significant enhancement after contrast. Additionally, there is an ovarian cyst located posterior to the uterus, presenting as a low T1, high T2 oval-shaped mass with a thin smooth wall; and a cervical nabothian cyst, with several low T1, high T2 signals visible in the cervix.

**2.Timing of Treatment:** The patient has no history of abortion in the past 3 months, no pelvic inflammatory disease, and no hearing or communication disorders. The MRI does not mention the presence of an intrauterine device (IUD), which must be confirmed before surgery, and if present, it should be removed 3 days before the procedure. The MRI does not mention any history of lower abdominal surgery or intestinal adhesions, so the patient's medical history must be carefully reviewed and assessed before surgery. There are no clear contraindications (pending confirmation of IUD and surgical history), and HIFU treatment can be considered.

**3.Treatment Objective and Doctor-Patient Communication:** The primary goal of the treatment is to use HIFU ablation technology to cause coagulative necrosis of the fibroid tissue, thereby reducing the fibroid volume, alleviating or eliminating symptoms caused by the fibroid, sauch as excessive menstruation and pressure symptoms, and reducing or avoiding surgery. In doctor-patient communication, it is important to explain the principles, indications, contraindications, expected outcomes, possible risks (such as skin burns, intestinal injury, nerve damage, etc.), and alternative treatment options for HIFU. It should also be emphasized that the current treatment is mainly for the fibroid on the left side of the uterus, while the ovarian cyst and cervical nabothian cyst usually do not require HIFU treatment unless there are special indications. Patients should be informed of the possibility of multiple treatments depending on the size and blood supply of the fibroid.

**4.Preoperative Preparation and Precautions:** Preoperative preparation includes confirming the absence of recent abortion, active pelvic inflammatory disease, and hearing or communication disorders. The presence of an IUD should be confirmed, and if present, it should be removed 3 days before surgery. The patient's history of lower abdominal surgery and possible intestinal adhesions should be assessed, with particular attention to scarring. The MRI report should be reviewed to confirm the location, size, and signal characteristics of the fibroid, as well as its relationship with surrounding structures. Preparation should include appropriate bladder filling and water sac tension, and common preoperative tests such as blood routine and coagulation function should be performed. Equipment, including the HIFU device and ultrasound contrast agents, should be checked to ensure they are in good working condition.

**5.Treatment Analysis and Intraoperative Precautions:** Due to the fibroid's size and rich blood supply, it is difficult to treat and may require multiple treatments. During treatment, pushing the anterior bowel may be necessary to obtain a clear acoustic window. Treatment should begin with the central area of the fibroid and gradually expand outward, following safety distances. The procedure should be performed layer by layer, monitoring real-time images and adjusting treatment strategies based on the fibroid's gray-scale changes. Ultrasound contrast agents may be used intraoperatively to assess blood perfusion and ablation effects.

**6.Postoperative Observation and Management:** Postoperatively, close monitoring of vital signs is required, with attention to any abdominal pain, fever, skin burns, or other complications. The urinary condition should be observed, as well as any abnormal vaginal bleeding or discharge. Symptomatic treatment should be provided for any minor discomforts such as abdominal bloating or low fever. Patients should be instructed on postoperative precautions, such as avoiding strenuous activities and maintaining genital hygiene. Follow-up appointments should be arranged at 1 month, 3 months, and 6 months, including symptom assessment, gynecological examination, and ultrasound or MRI to evaluate treatment effectiveness. Based on the results of the first treatment, decisions should be made about the need and timing for additional HIFU treatments. Any complications should be promptly addressed.

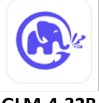

**GLM-4-32B**

Figure E6: Treatment Plan Generated by GLM

# F  EFFICIENCY ANALYSIS OF FUAS-AGENTS SYSTEM

|  | Planner Agent | Executor Agent | Strategy Agent | Optimizer Agent |
|---|---|---|---|---|
| Running Times | 1.3 | 29.43 | 66.70 | 18.73 |
| Token Usage (k) | 131 | 0 | 6334 | 3178 |
| Success Rate | 100% | 100% | 75% | 100% |

# G  VALUATION RESULTS FROM FOUR SENIOR FUAS CLINICIANS

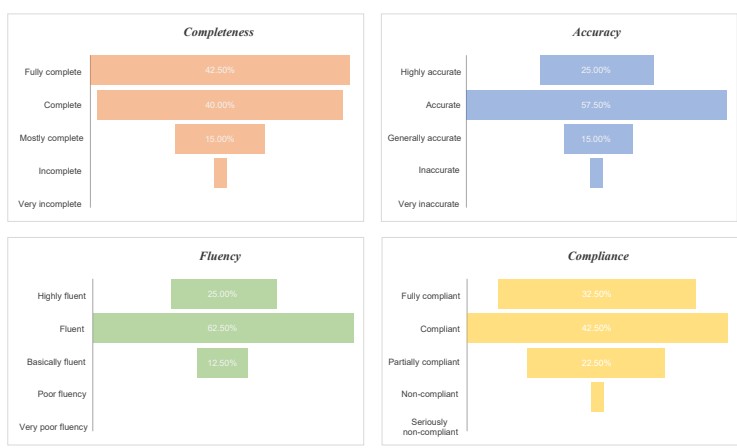

Figure G1: Valuation results from four senior FUAS clinicians

# H  LLM USAGE STATEMENT

We used large language models (LLMs) solely for minor language polishing, such as improving grammar and clarity of some sentences. No LLMs were involved in research ideation, experimental design, analysis, or drafting substantive content of this paper.

