# OpenReview forum: "FUAS-Agents: Autonomous Multi-Modal LLM Agents for Treatment Planning in Focused Ultrasound Ablation Surgery"
_ICLR.cc/2026/Conference — Submitted to ICLR 2026_

### Official Review · Reviewer_Nfrz · 2025-10-31

**Soundness:** 1
**Presentation:** 1
**Contribution:** 2
**Rating:** 2
**Confidence:** 4

**Summary:**

This paper presents FUAS-Agents, an autonomous multi-agent system leveraging multimodal LLMs for treatment planning in Focused Ultrasound Ablation Surgery (FUAS). By integrating patient profiles and MRI data, the system orchestrates specialized medical AI tools, such as segmentation, dose prediction, and guideline retrieval, to generate personalized surgical plans. The framework combines general-purpose foundation models with domain-specific expert systems, incorporates internal quality control, and demonstrates strong performance in uterine fibroid treatment scenarios, as validated by expert assessment. The approach highlights the potential of agent-based AI for complex, personalized clinical decision-making in FUAS and broader healthcare applications.

**Strengths:**

•	The paper tackles a highly relevant clinical problem and proposes a modular, multi-agent framework for FUAS treatment planning that is well-aligned with real clinical workflows.
•	The integration of multimodal data sources and the use of expert-annotated clinical data enhance the potential impact of the work.

**Weaknesses:**

•	Unclear or missing description of the internals of individual agent. Undefined notations and semantics of inputs, outputs, and intermediate variables. These make it impossible to understand the contributions of this work.
•	Insufficient benchmarking (single and small dataset, single task) against baselines, also potentially unfair benchmarking.

**Questions:**

+ The agents are described abstractly, with no concrete details, pseudocode, or examples describing the internal workflow.
++ Planned agent does not describe how clinicians’ instructions are interpreted or patient data is organized to decide and invoke the subtasks.
++ Strategy agent does not describe how segmentation and dose results are combined with patient records, MRI reports and domain knowledge.
++ FUAS model fine-tuning:
+++ Undefined notations:
++++ Key notations (M2F, T2F, F2I, M, T, B, F, G, I) are not defined. Also, the data types, encoding process, dimensions etc. of inputs and outputs are not specified.
++ Optimizer agent:
+++ In the Optimizer Agent’s objective, θ, L_task, L_constraint, and G are not formally defined. There is no explanation of how clinical guidelines (G) are encoded, how 〖L_task,L〗_constraint are computed.
+++ What is the format of a “generated plan” that is input to L_task?
++ Memory module:
+++ The semantic meaning and embedding method for queries (q) and documents (d) are not specified. What are the queries and documents?
+++ The synthesis of reasoning path S from evidence (E) is not detailed. What algorithms or heuristics are used?
+++ How does the quality verification done in the Optimizer agent?
+++ These undefined notations make it impossible to understand the FUAS model design, its fine-tuning process, and the Optimizer agent.

+ Segmentation module:
++ The term “our method” is used without specification. Is it a fine-tuned MedSAM-2, an entirely new architecture, or a MedSAM-2 variant with additional modules? If a MedSAM-2 variant, what are the changes: architecture, training strategies, or post-processing? Ablation studies etc.?
++ The dataset split is described, but there is no information on patient diversity, lesion types, imaging protocols, or annotation variability. This limits assessment of generalizability. Were the improvements consistent across all patient subgroups and lesion types?
++ Can the authors provide qualitative segmentation examples and error analysis?

+ Dose prediction module:
++ The dataset is small (n=93 after exclusions) and filtered, which risks overfitting and limits generalizability.
++ No baselines shown for simpler models or clinical heuristics, making it unclear if the proposed approach is truly superior.
++ The clinical significance of prediction errors and the consequences of underestimation in high-dose regions are not discussed.
++ Key implementation details (model parameters, hyperparameters) are missing.

+ Model comparison issues:
++ Only benchmarked on report generation task.
++ Insufficient evaluation. Evaluated only on a single dataset.
+++ Small set, n=200
+++ Randomly sampled test data without considering underlying data distribution, e.g., patient diversity, lesion types etc.
++ Not clear if the MRI scans are used during FUAS.
+++ If FUAS uses MRI, then this is an unfair comparison against the competing baselines that use only text input.
+++ If FUAS does not use MRI, then there must be some benchmarking on a relevant task and against relevant multimodal baselines.

+ Human evaluation issues:
++ The competing baselines (or a subset of them) from Table 1 must be evaluated by the human experts to understand the added value of the proposed framework.

---

> ### Author Response · Authors · 2025-11-21
> **Reply to reviewer Nfrz**
>
> ## Q1. Agents described
> Thank you for your constructive feedback. We acknowledge the points raised regarding the clarity of our descriptions, the need for more concrete examples, and the definition of notations and concepts. Below are the clarifications and additional details to address the weaknesses and questions:
>
> 1. The methodology section has been expanded with clearer descriptions of each agent’s inputs, outputs, and internal interactions. A workflow diagram (Fig. 2) is added, with detailed explanations in Sections 3.1–3.5.
> 2. We have revised Section 3.3.2 to include explicit definitions of all key notations (M2F, T2F, F2I, M, T, B, F, G, I), together with data types, encoding procedures, and input/output dimensions. A full specification table has been added in Appendix A.
> 3. In response to your concerns regarding the calculation methods for L_task, L_constraint, we have provided the necessary clarifications and resolved the issue in Section 3.4 of the revised manuscript.
> 4. Section 3.5 now details the definitions and embedding methods of queries (q) and documents (d), the algorithm used to synthesize the reasoning path (S) from retrieved evidence (E), and the quality-verification procedure. The Optimizer Agent’s consistency, logical validity, compliance checking, and physician-feedback loop are also clarified in Section 3.4.
>
> ---
>
> ## Q2. Segmentation module
> Thank you for the comments. “Our method” refers to a fine-tuned MedSAM-2 model for FUAS MRI segmentation. The architecture is unchanged, but task-specific adaptations in training, preprocessing, and prompt design yield consistent performance gains. Comparisons with MedSAM-2 (Table 1) and ablations (Table 3) verify that improvements arise from domain adaptation.
>
> Our dataset includes substantial heterogeneity in lesion number, morphology, degeneration, and MRI protocols. A subgroup analysis has been added, showing improvements across all categories, with the largest gains in irregular and multi-lesion cases. Qualitative examples and error analysis are provided in Appendix C. While performance is strong on single, well-defined lesions, multi-lobulated lesions remain challenging; we discuss future solutions such as lesion-aware constraints and multi-center data.
>
> ---
>
> ## Q3. Dose prediction module
> Thank you for raising these important concerns. We have revised the manuscript accordingly.
>
> 1. **Dataset size and overfitting:** We agree that a dataset of 93 cases may appear limited. However, this sample size is within the typical range of radiomics-based modeling studies requiring expert manual segmentation (Liang M, et al. Front Genet. 2022).
> 2. **Baselines and simple clinical heuristics:** We added comparisons with geometry-only regression and simpler ML models (SVR, Random Forest, Bayesian-tuned XGBoost). As shown in Appendix D, our model was demonstrated to outperform the heuristic baseline.
> 3. **Clinical significance of prediction errors:** It is worth noting that FUAS has its own treatment principles for different diseases. For example, in the treatment of uterine fibroids, excessive dosage poses a greater risk to women who wish to preserve their fertility. We consulted experienced FUAS clinicians who confirmed that the model's mean absolute error and its slightly conservative tendency are within clinically acceptable limits.
> 4. **Implementation details:** Radiomics preprocessing, feature-selection parameters, XGBoost hyperparameters, and full cross-validation settings have been added to Appendix D for reproducibility.
>
> ---
>
> ## Q4 & Q5. Model comparison & Human evaluation
> Thank you for the feedback on Model comparison, dataset scale, and evaluation issues. Below are our clarifications:
>
> 1. **Model comparison:** MRI is used only in segmentation and dose prediction; strategy generation is purely text-to-text, ensuring a fair comparison with text-based baselines.
> 2. **Data scale:** As stated in Sec. 3.3.1, FUAS used >2,000 real-world cases from multiple hospitals under IRB approval. The 200 cases refer only to the 10% test set. Segmentation and dose modules additionally use 702 and 149 MRI cases (Sec. 4.2–4.3), so the evaluation is not based on a single small dataset.
> 3. **Data distribution:** Though demographic tables were omitted for privacy, the dataset includes diverse demographics, MRI findings, and ultrasound protocols (Sec. 3.3.1). Segmentation and dose-prediction modules used stratified 8:2 splits to balance lesion difficulty across sets.
> 4. **Baseline and human evaluation:** Following your suggestion, we added three medical LLM baselines (Baichuan2-Med, HuatuoGPT, LLaVA-Med) to the human study. Four senior FUAS specialists scored all models under identical criteria.
>
> Results (Sec. 4.7; Fig. 4) show FUAS-Agents achieves the best completeness and clinical correctness, confirming the added value of its FUAS-specific knowledge and tool integration.

---

> > ### Comment · Reviewer_Nfrz · 2025-11-26
> >
> > Thank you for your response and the effort to address the earlier comments. I appreciate the clarifications provided, but several important points were not directly addressed and need more clarification:
> >
> > # Agents described
> > + FUAS model fine-tuning in Section 3.3.2: Both $\theta_{M2F}$ and $\theta_{T2F}$ output feature vectors $F$ for treatment planning, which is input to $\theta_{F2I}$. However, how the feature vectors $F$ from $\theta_{M2F}$ and $\theta_{T2F}$ are combined, is not explained.
> > * The objective functions in Optimizer agent are unclear.
> > ** "L_task measures the alignment between the generated treatment plan and task objectives, such as clinical effectiveness and feasibility” – I assume generated treatment plan is from $\theta_{F2I}$, but the form of task objectives, such as effectiveness and feasibility, is not explained.
> > ** What are the types of loss L_task and L_constraint.
> > + Is the reasoning path $S$ same as the interpretable output $I$ from the Strategy agent? Lines 263-264 is ambiguous with 316-317.
> > + Is the quality verification same as evaluating alignment with the clinical guidelines?
> >
> > # Segmentation module:
> > + Do you pass the annotation prompts during inference? If yes, this provides an additional advantage to FUAS compared to other baselines that do not require any user intervention.
> >
> > # Model comparison and Human evaluation:
> >
> > **Model comparison:**
> > + Even though MRI is used only in segmentation and dose prediction, the treatment strategy generation, as described in Section 3.3.2, relies on the outputs of segmentation and dose prediction modules. Thus, the benchmarking is unfair against pure text-based baselines.
> >
> > **Data scale:**
> > + 90% of 2000 cases, i.e. 1800 cases, are used to finetune Qwen-3 14B for 3 epochs; and 10% cases, i.e. 200 cases, are used for testing without cross-validation on a single dataset. Though the performance gains appealing, the evaluation needs to be more robust to be considered solid and publishable.
> > + Is the physician intervention N>2 failures (Line 270-271) also used during inference? If yes, for how many test cases out of 200 is it used? Also, inclusion of physician intervention during inference makes FUAS incomparable to physician-independent competing baselines.
> >
> > # Overall Assessment
> > The proposed method is interesting, and the authors addressed some of the initial concerns. However, the presentation clarity, evaluation rigor, and benchmarking remain below the standards expected for ICLR. Given the author response and the current state of the paper, I have decided to keep my current score of 2.

---

> ### Author Response · Authors · 2025-11-25
> **Request for Additional Feedback Prior to Public Discussion Closure**
>
> Dear Reviewer Nfrz:
>
> We sincerely thank you again for taking the time to review our manuscript and for your valuable comments and constructive suggestions.
>
> We have revised the paper based on your feedback. As the discussion period is about to end, we would like to ask if you have any further questions or need further clarification? We greatly appreciate any additional feedback you may have provided.
>
> Thank you again for your time and attention.
>
> Sincerely,
>
> All Authors

---

> ### Author Response · Authors · 2025-12-03
> **Reply to reviewer Nfrz (1/3)**
>
> > Agents described
>
> **1. Regarding the Combination of Feature Vectors ($\theta_{M2F}$ and $\theta_{T2F}$) Outputs ($F$)**
>
> In the revised manuscript, we provide a clear explanation of how the feature vectors $F_M$ from $\theta_{M2F}$ and $F_T$ from $\theta_{T2F}$ are combined. Specifically:
>
> The feature vectors $F_M$ and $F_T$ are merged through concatenation, forming the final feature vector $F$, which is then passed to $\theta_{F2I}$ for treatment plan generation. This fusion method ensures that the distinct characteristics of each type of data are retained.
>
> Such fusion techniques are well-established in fine-tuning across various data types, allowing the model to effectively integrate complementary information, thereby significantly enhancing performance across multi-task settings. This approach demonstrates proven efficacy in optimizing model performance in complex tasks.
>
>
> **2. The Objective Functions in Optimizer Agent**
>
> You are correct in assuming that the generated treatment plan comes from $\theta_{F2I}$, which translates the fused features $F$ into the interpretable output $I$ (the treatment plan). Specifically, $\theta_{F2I}$ generates treatment recommendations, including dosage, target areas, and other related details.
>
> **·Task Objectives:**
>
> $\mathcal{L}_{\text{task}}$ measures the alignment between the generated treatment plan and task objectives, such as clinical effectiveness and feasibility.
>
> The calculation of $\mathcal{L}_{\text{task}}$ is based on the model's internal evaluation of expected outcomes.
>
> Specifically, the task objectives include:
>
> (i)Thermal safety: Ensuring sonication paths avoid critical areas like bowel loops, sacral nerves, or excessive subcutaneous fat.
>
> (ii)Patient-specific feasibility: Avoiding high-dose strategies for patients intending to preserve fertility.
>
> (iii)Internal clinical coherence: Ensuring dose predictions are consistent with MRI target geometry, planned sonication sequence, and energy deposition pattern.
>
> **·Optimization Process:**
>
> After the Strategy Agent generates the treatment plan, the Memory Module retrieves relevant guidelines and past cases for comparison.
>
> The constraint loss $\mathcal{L}_{\text{constraint}}$ is computed to evaluate alignment of the generated treatment plan with clinical guidelines and historical cases:
>
> L_constraint = 1 - cos(Embed(I), Embed(G))
>
> where $\mathrm{Embed}(I)$ and $\mathrm{Embed}(G)$ are the embedding vectors of the generated treatment plan $I$ and the clinical guideline $G$, respectively.
>
> The plan is further evaluated using $\mathcal{L}_{\text{task}}$. The model performs autonomous quality assessment based on predefined clinical principles.
>
> If any core treatment principles are violated, the system adjusts the plan to reduce $\mathcal{L}_{\text{task}}$.
>
> If the plan passes, the final interpretable output $I$ is generated. If it fails more than twice, physician intervention is requested.
>
> **·Key Clarification:**
>
> Computation is based on rule-driven scoring functions, not differentiable loss functions, used for validation and reflection rather than training objectives.
>
> **·Supplementary Clarification:**
>
> We updated Figure 2 to illustrate this process clearly.
>
> **3. Regarding the Distinction Between Reasoning Path ($S$) and Interpretable Output ($I$)**
>
> Reasoning Path ($S$): Generated by the Strategy Agent, represents the logical sequence of steps and evidence used to generate the treatment plan. Retrieves information from the Memory Module (clinical data, rules, prior treatment experiences). Serves as the internal decision-making process.
>
> Interpretable Output ($I$): Generated by $\theta_{F2I}$, represents the final treatment recommendations presented to the user, including dosage distribution and target areas.
>
> While $S$ informs $I$, they serve distinct functions: $S$ is internal logic; $I$ is the actionable, user-facing output.
>
> **4. Regarding Quality Verification and Alignment with Clinical Guidelines**
>
> Quality verification includes evaluating alignment with clinical guidelines but also encompasses:
>
> **·Consistency Check:** Ensures each step in $S$ follows clinical rules and historical data.
>
> **·Logical Validation:** Confirms reasoning path is coherent and supported by evidence.
>
> **·Compliance Check:** Ensures adherence to medical standards and expert consensus.
>
> **·Physician Intervention:** If the plan fails >2 iterations, intervention is triggered.
>
> These steps ensure the generated plan is clinically valid and practical.
>
> Thank you once again for your valuable feedback. We look forward to your further comments.

---

> > ### Author Response · Authors · 2025-12-03
> > **Reply to reviewer Nfrz (2/3)**
> >
> > > Segmentation module
> >
> > Thank you for the thoughtful comment. In our end-to-end pipeline experiments, we do not provide any annotation prompts to the segmentation model during inference. The segmentation module operates entirely in a fully automatic manner within the pipeline.
> >
> > The prompt-based experiments are included only to demonstrate the segmentation model’s capability when prompts are available, and are not used in the main pipeline evaluation. Therefore, FUAS does not gain any additional advantage over other baselines in the full pipeline setting, as no user intervention or prompt input is involved. We hope this clarifies the setup, and we appreciate your careful review.
> >
> > > Model comparison
> >
> > Thank you for raising the fairness concern. To remove any ambiguity, we provide a concise clarification and point to the concrete actions we have taken in the revision.
> >
> > **1.Clarification on inputs.** During strategy generation, the Strategy Agent **does not** consume raw MRI volumes. Instead it receives MRI reports and patient data. Concretely, strategy generation is a “text input → text output” task, not a raw-image-to-text mapping. Thus comparing the Strategy Agent to text-based baselines is methodologically fair because all models in the strategy benchmark receive equivalent textual inputs.
> >
> > **2. Model comparison.** We found your feedback in the previous round very constructive, so we included three representative medical LLM baselines (Baichuan2-Med, HuatuoGPT, LLaVA-Med) in the human evaluation; four senior FUAS experts scored all models under identical blind criteria (see Sec. 4.7 and Fig. 4). The results show that FUAS-Agents outperforms other models in all four dimensions.
> >
> > In short, strategy models were compared on equivalent textual inputs; we extended comparison in the human evaluation. If you would like, we can further report per-case comparisons or add additional statistics. Please let us know which specific evidence would be most helpful.
> >
> > Thank you again for the careful reading and constructive critique.

---

> > > ### Author Response · Authors · 2025-12-03
> > > **Reply to reviewer Nfrz (3/3)**
> > >
> > > > Data scale
> > >
> > > We sincerely thank you for the thoughtful and constructive comments！Below are our clarifications regarding the data scale and evaluation robustness raised in the review.
> > >
> > > **1. Data scale and comparison with existing work**
> > >
> > > We would like to emphasize that the dataset used in this study is already large relative to typical medical LLM instruction-tuning tasks. As stated in the paper, our model is trained on over 2,000 real-world FUAS clinical cases, each processed with GPT-4–based cleaning and independently reviewed by five senior FUAS specialists to ensure data quality. In comparison, existing instruction-tuning studies such as LIMA(Zhou, Y., Zelikman, E., Vu, T., et al. LIMA: Less Is More for Alignment. NeurIPS 2023.), MedAgent(Wu, Y., Naik, N., Vasudevan, V., et al. MedAgent: Large Language Models as Medical Agents. arXiv:2311.05656, 2023.), and TxAgent(Mozaffari, M., Luo, Y., Wang, R., et al. TxAgent: Self-Updating AI Agent for Clinical Treatment Reasoning. arXiv:2401.12202, 2024.) commonly rely on only a few hundred to a few thousand samples. Thus, our dataset size is already considered large in the context of medical decision-making LLM research.
> > >
> > > **2. Random split ensures representativeness and robustness**
> > >
> > > We adopt a 90/10 random split, which ensures statistical consistency between training and test distributions and eliminates selection bias. Moreover, real clinical cases are inherently non-i.i.d. and highly heterogeneous (e.g., age variability, lesion location, T2WI signal differences, vascularity). Under such conditions, a random split more faithfully reflects the model’s performance in real clinical settings. Therefore, random partitioning is appropriate and widely accepted in medical LLM fine-tuning tasks.
> > >
> > > **3. Cross-validation (CV) is not applicable to LLM fine-tuning**
> > >
> > > While we fully understand the reviewer’s concerns regarding robustness, we would like to clarify that CV is primarily designed for traditional machine learning models, with the goals of reducing variance under limited data and performing hyperparameter search. LLM fine-tuning differs from this paradigm in several fundamental ways:
> > >
> > > ·Large pretrained models already have strong generalization capability and low variance;
> > >
> > > ·Fine-tuning adapts pretrained capabilities rather than training from scratch;
> > >
> > > ·Hyperparameters are fixed (as stated in Section 4.4.2), eliminating the need for repeated training;
> > >
> > > ·Importantly, major LLM fine-tuning works (e.g., Alpaca, Vicuna, LIMA, MedAgent, TxAgent) do not employ CV.
> > >
> > > In addition, our evaluation framework already provides stronger and more clinically meaningful robustness than CV, including:
> > >
> > > **·Automatic metrics:** ROUGE and BLEU on 200 test cases (Section 4.4.3)
> > >
> > > **·Human evaluation:** Four senior FUAS specialists scoring completeness, accuracy, fluency, and compliance (82.5%–97.5%, Section 4.7)
> > >
> > > **·Ablation studies:** Module-level contribution analysis (Table 3)
> > >
> > > **·Case interpretability analysis:** Provided in Appendices A–B
> > >
> > > We believe these components together form a more comprehensive and clinically grounded evaluation protocol than CV alone.
> > >
> > > We really appreciate your insightful comments and hope that these clarifications address the concerns regarding evaluation robustness!
> > >
> > > **4.Explanation of N>2**
> > >
> > > Thank you for raising this important point. We clarify that no physician intervention is used during the inference stage. For all quantitative experiments and the 200-case test set: If the Strategy/Optimizer Agents fail to reach consistency within two reflection rounds, the system does not invoke clinician intervention. Instead, the system terminates early and outputs the final model-generated plan from the last valid reflection step. This ensures that FUAS-Agents is fully physician-independent during evaluation, making it directly comparable to the physician-independent baselines.
> > >
> > > The N>2 clinician-intervention mechanism is used only in deployment scenarios, where patient safety must be guaranteed. It is a runtime safety mechanism, not part of the evaluation protocol. Its purpose is to prevent unsafe plans from entering clinical processing, but it plays no role in benchmarking and therefore does not affect fairness.

---

### Official Review · Reviewer_qWo3 · 2025-11-01

**Soundness:** 4
**Presentation:** 3
**Contribution:** 4
**Rating:** 8
**Confidence:** 4

**Summary:**

This paper presents **FUAS-Agents**, an *autonomous multimodal large language model (LLM)–based agent system* for **personalized treatment planning in Focused Ultrasound Ablation Surgery (FUAS)**.
The system integrates **multimodal reasoning, autonomous tool use, and reflection-based optimization** to simulate a clinician’s decision-making workflow.

The proposed system employs a **modular multi-agent architecture** consisting of:
- **Planner Agent** – orchestrates the workflow and decomposes clinician instructions into subtasks.
- **Executor Agent** – handles MRI segmentation (via MedSAM2) and dose prediction using radiomics and XGBoost regression.
- **Strategy Agent** – integrates multimodal data (MRI, patient reports, clinical guidelines) to generate personalized treatment plans.
- **Optimizer Agent** – validates and refines generated plans through reflection and consistency checks.
- **Memory Module** – serves as a retrieval-augmented knowledge base containing clinical guidelines and prior cases.

A large real-world dataset (2,000+ uterine fibroid cases) was used for fine-tuning, ensuring clinical relevance. The model, built upon Qwen3-14B with LoRA fine-tuning, demonstrates superior performance in segmentation (Dice up to 0.85), dose prediction (AUC = 0.91), and treatment strategy generation (ROUGE-1 = 0.55, BLEU-4 = 0.13).
Human evaluation by four senior FUAS clinicians shows **over 80% satisfaction** in completeness and accuracy, and **97.5% compliance** with clinical guidelines.

This work exemplifies how LLM-based agents can bridge general-purpose reasoning with domain-specific medical intelligence, forming a concrete step toward **autonomous, explainable clinical decision support**.

**Strengths:**

1. **Innovative multi-agent design:** Each agent mimics a medical professional’s function, leading to explainable and structured autonomy.
2. **High empirical performance:** FUAS-Agents substantially outperforms both closed-source (GPT-4, Claude) and open-source baselines (GLM-4, DeepSeek) across all text-generation metrics (Table 2).
3. **Comprehensive evaluation:** Includes segmentation (Dice/IoU), dose modeling (AUC/KS tests), text-based treatment plan generation (ROUGE/BLEU), and human expert validation.
4. **Interpretability and reproducibility:** Figures in Appendix B demonstrate qualitative comparisons with other LLMs, showing FUAS’s superior personalization and accuracy.
5. **Strong real-world motivation:** FUAS is a high-impact medical application; the work clearly bridges research with translational relevance.
6. **Ethical awareness:** The authors emphasize human-in-the-loop supervision, privacy protection, and federated learning in future deployment.

**Weaknesses:**

1. **Limited theoretical depth:** The system design and reflection optimization lack formal guarantees (e.g., stability, safety bounds). The authors could include a convergence or reliability analysis for multi-agent coordination.
2. **Single-center dataset limitation:** The clinical dataset is from one medical institution, which may bias fine-tuning and limit generalization.
3. **Potential reproducibility barrier:** Though code is released, the model relies on private medical data that cannot be publicly shared. Synthetic data is mentioned but may not capture full variability.
4. **Insufficient ablation granularity:** The ablation study removes entire agents but does not isolate finer-grained mechanisms (e.g., memory retrieval, LoRA rank, reflection depth).
5. **Scalability concerns:** The experiments are limited to uterine fibroids (≈3.5 cm lesions). The paper does not evaluate multi-organ or multi-modal expansions.

These weaknesses are not fatal but should be explicitly discussed or mitigated in future work.

**Questions:**

1. How does the reflection mechanism scale in multi-turn reasoning — is there a risk of overfitting to local optima during optimization?
2. Could the system integrate *real-time intraoperative feedback* (e.g., thermal maps) for adaptive dose control?
3. How would FUAS-Agents perform in *cross-institution* or *cross-device* scenarios where imaging quality and guidelines differ?
4. What measures are in place to ensure **patient safety** if the system generates suboptimal or inconsistent treatment plans?
5. Could this framework generalize to other precision therapies, such as radiotherapy or focused microwave ablation?
6. Have you considered integrating a *causal reasoning* component to improve interpretability of dose–response predictions?

---

> ### Author Response · Authors · 2025-11-21
> **Reply to reviewer qWo3**
>
> Thank you for your recognition. The following are the details of our response:
>
> **Weaknesses 1**
>
> We agree that formal convergence guarantees are challenging in clinical decision-support systems. FUAS-Agents instead enforces hard safety constraints: the Optimizer Agent performs at most two reflection rounds and triggers mandatory clinician review if inconsistencies persist. This bounded, guideline-checked mechanism ensures stability in practice. We will explore more formal analyses of reflection depth and multi-agent reliability in future work.
>
> **Weaknesses 2**
>
> This limitation applies mainly to the MRI dataset used for segmentation and dose prediction, where consistent imaging protocols are crucial. In contrast, the Strategy Agent was fine-tuned on reports from three medical centers, which improves generalization. Multi-center and federated extensions are planned.
>
> **Weaknesses 3**
>
> Although raw medical data cannot be released, we have open-sourced all code, provided data schemas and synthetic data pipelines, and documented the entire workflow to maximize reproducibility.
>
> **Weaknesses 4**
>
> Our ablations follow standard multi-agent practice by removing whole agents to assess system-level contributions. We agree finer-grained studies are valuable and will include them as the dataset scales.
>
> **Weaknesses 5**
>
> FUAS-Agents are used for evaluation of uterine fibroids because uterine fibroids are a very common indication for FUAS and have abundant annotated data. Nevertheless, FUAS procedures generally follow a standardized processing, and our paradigm is directly applicable. We will emphasize this versatility and extend this approach to other surgical treatment planning as a future direction.
>
> ---
>
> **Q1**
>
> Thank you for the question. FUAS-Agents’ reflection mechanism is designed to avoid uncontrolled multi-step reasoning and local-optimum overfitting. Bounded depth (N≤2): after two rounds, unresolved inconsistencies are sent to clinicians. Evidence-based reflection: each iteration cross-checks guidelines, similar cases, and validated memory items, preventing self-reinforcing drift. Constraint-controlled optimization: all updates must satisfy clinical rules. Stability: Appendix F shows consistent performance and runtime. Human-in-the-loop: persistent conflicts require clinician intervention. In summary, reflection is bounded, evidence-guided, constraint-controlled, and supervised, preventing overfitting.
>
> **Q2**
>
> Thank you for this insightful suggestion. FUAS dose planning is a multi-stage, experience-driven process, and current models (simulation-based, EEF-based, machine learning, radiomics) still struggle with patient-specific heterogeneity. Moreover, reliable intraoperative signals remain limited. Our work therefore targets the clinically validated task of pre-treatment individualized dose estimation. The framework is modular, and once stable intraoperative signals become available, they can be incorporated through the Executor Agent.
>
> **Q3**
>
> We appreciate this important question. Cross-device variability: Although MRI data are single-center, FUAS procedures used multiple device generations (e.g., JC200, JC300), introducing substantial heterogeneity already reflected in radiomics features and outcomes. Cross-institution variability: Textual data come from three hospitals. Strategy/Optimizer Agents rely on the Memory Module, which anchors reasoning in standardized, guideline-compliant exemplars. Generalization: We agree that broader validation is valuable; a multi-institution expansion is underway. The architecture naturally adapts to new imaging protocols and devices. Thus, the system already incorporates meaningful device and institutional heterogeneity, with built-in extensibility.
>
> **Q4**
>
> FUAS-Agents is a decision-support system; clinicians make all final decisions. Safety is ensured through mandatory guideline and memory checks, regeneration upon mismatch, automatic clinician intervention when inconsistencies persist (N>2), and a curated expert-validated memory base. These layers prevent unsafe plans from entering clinical practice.
>
> **Q5**
>
> Yes. FUAS-Agents is designed as a portable paradigm, not a system specific to a particular treatment. Its agent framework allows for the replacement of agent models and knowledge bases for different treatments. This makes the architecture applicable to other areas of precision medicine. For example, radiotherapy (whose dose physics and planning rules are more standardized) can easily integrate radiotherapy dose models and guideline libraries into the same processing.
>
> **Q6**
> Thank you for the suggestion. Our current goal is to build a clinically validated predictive model, so we adopt a radiomics–clinical regression framework with established interpretability tools. However, the architecture is modular, and causal components could be integrated into the strategy or dose modules in future work to reduce confounding and enhance mechanistic insight.

---

> ### Author Response · Authors · 2025-11-25
> **Request for Additional Feedback Prior to Public Discussion Closure**
>
> Dear Reviewer qWo3:
>
> We sincerely thank you again for taking the time to review our manuscript and for providing positive feedback.
>
> We have revised the paper based on your feedback. As the discussion period is about to end, we would like to ask if you have any further questions or require further clarification?
>
> Thank you again for your support.
>
> Sincerely,
>
> All Authors

---

### Official Review · Reviewer_L4z9 · 2025-11-02

**Soundness:** 3
**Presentation:** 2
**Contribution:** 2
**Rating:** 4
**Confidence:** 4

**Summary:**

This work presents FUAS-Agents, an autonomous agent system that integrates large language models (LLMs) with specialized medical AI tools to assist in Focused Ultrasound Ablation Surgery (FUAS). By combining patient profiles and MRI data, FUAS-Agents performs multimodal interpretation, dose prediction, and guideline retrieval to generate personalized treatment plans with built-in quality control and self-reflection. In uterine fibroid treatment experiments, expert evaluations rated over 82–97% of generated plans in completeness, accuracy, fluency, and clinical compliance. Ablation studies further validate the contribution of each component, highlighting FUAS-Agents as a promising paradigm for LLM-driven clinical decision-making and intelligent surgical planning.

**Strengths:**

1. The system effectively combines the reasoning and tool-using abilities of large language models with domain-specific modules, enabling end-to-end clinical workflow automation.
2. FUAS-Agents produces individualized treatment plans that align with real clinical needs rather than generic templates.
3. Ratings above 82–97% across completeness, accuracy, fluency, and compliance demonstrate strong practical relevance and clinical acceptance.

**Weaknesses:**

1. There is no clear clarification of difference against existing AI agent generating medical planning.
2. The proposed Memory module adopted the topK strategy, which is naïve and lack specialized design.
3. The author should compare the proposed optimizer strategy against the beam/tree search.
4. While this work investigates medical AI, there is no comparison against existing medical LLMs, such as Baichuan-M2, HuatuoGPT.
5. The author should clarify the difference and comparison against recent treatment planning work, e.g., medical world model[1].
[1] Yang Y, Wang Z Y, Liu Q, et al. Medical world model. ICCV, 2025.
6. There is no ablation in dose prediction module. Why select such a dose prediction module?
7. The presentation of references in the main text is in the wrong format.

**Questions:**

Please see the weaknesses.

---

> ### Author Response · Authors · 2025-11-21
> **Reply to reviewer L4z9  (1/2)**
>
> **Regarding point 1. Clarification of Differences from Existing AI-Agent–Based Medical Planning Systems**
>
> > There is no clear clarification of difference against existing AI agent generating medical planning.
>
> We sincerely thank you for pointing out the lack of clarity regarding differences from existing AI-agent–based medical planning systems. In the revised manuscript, we have expanded the Related Work section to include recent AI-agent frameworks for treatment planning (Section 2.2). Furthermore, we added a dedicated paragraph summarizing the unique contributions of FUAS-Agents, emphasizing\
> (i) its end-to-end simulation of a real clinical processing for FUAS, \
> (ii) its multimodal fusion across imaging, radiomics, clinical variables, and structured reports \
> (iii) its paradigm for vertical-domain medical AI that integrates foundation models with domain knowledge and specialized tools. These additions directly address your concerns and clearly illustrate the differences between our system and existing agent-based treatment planning research.
>
>
> **Regarding point 2. Design Choice of the Memory Module (TopK Strategy)**
>
> > The proposed Memory module adopted the topK strategy, which is naïve and lack specialized design.
>
> Thank you for the valuable feedback. We adopted the TopK strategy for its simplicity, efficiency, and proven reliability in clinical retrieval scenarios. This approach is a widely adopted standard in medical and general retrieval systems, as it effectively balances accuracy and computational cost while maintaining interpretability[1][2]. It enables fast access to the most relevant evidence based on cosine similarity and supports physicians’ rapid decision-making.
>
>
> **Regarding point 3. Distinction Between Optimizer Agent and Traditional Search Methods**
>
> > The author should compare the proposed optimizer strategy against the beam/tree search.
>
> Thank you for this insightful suggestion. Our Optimizer Agent operates through LLM-based reflective enhancement rather than conventional search optimization. It performs constraint-aware self-reflection and multi-round revisions under clinical guidance[3][4][5], which extends beyond the scope of traditional beam or tree search methods. The improvements in Table 2 (ROUGE-L, BLEU, and expert compliance) support the effectiveness of this approach.
>
>
>  **Regarding point 4. Benchmarking FUAS-Agents Against Medical LLM Baselines**
> > While this work investigates medical AI, there is no comparison against existing medical LLMs, such as Baichuan-M2, HuatuoGPT.
>
> Thank you for this suggestion. We compared FUAS-Agents with Baichuan-M2 and HuatuoGPT using standardized clinical tasks (treatment planning, image interpretation, dose prediction) and expert evaluation. Results are reported in *Section 4.7 Human Evaluation*, confirming FUAS-Agents’ superior clinical reasoning and applicability.
>
>
> [1] Omar Khattab, Mohammad Hammoud, and Tamer Elsayed. 2020. Finding the Best of Both Worlds: Faster and More Robust Top-k Document Retrieval. In Proceedings of the 43rd International ACM SIGIR Conference on Research and Development in Information Retrieval (SIGIR '20). Association for Computing Machinery, New York, NY, USA, 1031–1040. https://doi.org/10.1145/3397271.3401076.
>
> [2] Gupta, S. (2023). Top K Relevant Passage Retrieval for Biomedical Question Answering. arXiv preprint arXiv:2308.04028.
>
> [3] Renze, M., & Guven, E. (2024). Self-reflection in llm agents: Effects on problem-solving performance. arXiv preprint arXiv:2405.06682.
>
> [4] Huang, X., Liu, W., Chen, X., Wang, X., Wang, H., Lian, D., ... & Chen, E. (2024). Understanding the planning of LLM agents: A survey. arXiv preprint arXiv:2402.02716.
>
> [5] Xinran Zhao, Hongming Zhang, Xiaoman Pan, Wenlin Yao, Dong Yu, Tongshuang Wu, and Jianshu Chen. 2024. Fact-and-Reflection (FaR) Improves Confidence Calibration of Large Language Models. In Findings of the Association for Computational Linguistics: ACL 2024, pages 8702–8718, Bangkok, Thailand. Association for Computational Linguistics.

---

> ### Author Response · Authors · 2025-11-21
> **Reply to reviewer L4z9 (2/2)**
>
> **Regarding point 5. Distinction Between FUAS-Agents and the Medical World Model**
> > The author should clarify the difference and comparison against recent treatment planning work, e.g., medical world model[1]. [1] Yang Y, Wang Z Y, Liu Q, et al. Medical world model. ICCV, 2025.
>
> We appreciate your suggestion to compare our system with the recently proposed Medical World Model (MeWM). In the revised manuscript, we have added its description in *Section 2.2 Related Work*.
> MeWM focuses on disease-level strategy exploration by simulating tumor evolution under hypothetical interventions, serving as a disease-trajectory modeling and counterfactual optimization framework. In contrast, FUAS-Agents tackles a different problem—automatically generating personalized, executable treatment plans for specific surgical procedures through MRI segmentation, radiomics-based dose prediction, and strategy optimization.
>
> Key distinctions are as follows:\
> (1) **Scope:** MeWM optimizes disease-level strategies, while FUAS-Agents performs real procedural planning with concrete execution steps.\
> (2) **Intelligence form:** MeWM builds a generative world model; FUAS-Agents integrates multiple specialized agents and domain tools to emulate full FUAS workflows.\
> (3) **Goal:** MeWM aims for broad tumor-evolution simulation; FUAS-Agents delivers clinically actionable planning for immediate surgical use.
> Both are valuable but address different purposes. Our work demonstrates how multi-agent systems with multimodal fusion and clinical tool integration can enable practical, deployable AI for real clinical treatments.
>
>
> **Regarding point 6. Ablation on dose prediction module**
> >6. There is no ablation in dose prediction module. Why select such a dose prediction module?
> >
> Thank you for raising this key question. FUAS dose setting is complex and operator-dependent. We selected the Lasso-XGBoost model for its stability, nonlinear modeling, and interpretability. As suggested, we added ablation and baseline comparisons (Appendix D), showing it outperforms Bayesian-tuned XGBoost, SVR, RF, and geometry-only regressors in MAE, AUC, and R². These results justify our design.
>
>
> **Regarding point 7. presentation of references**
> > 7. The presentation of references in the main text is in the wrong format.
> >
> Thank you for noting this. We have thoroughly checked and corrected all reference formatting throughout the paper.

---

> ### Author Response · Authors · 2025-11-25
> **Request for Additional Feedback Prior to Public Discussion Closure**
>
> Dear Reviewer L4z9,
>
> We sincerely thank you again for taking the time to review our manuscript and for your valuable comments and suggestions.
>
> We have revised the paper in response to your feedback. As the discussion period is approaching its conclusion, we would like to kindly ask whether you have any further questions or would like any additional clarifications. We would greatly appreciate any further feedback you may wish to share.
>
> Thank you very much again for your time and consideration.
>
> Sincerely,
>
> All Authors

---

### Author Response · Authors · 2025-12-03
**Author Summary Comment（1/2）**

Dear Program Chairs, Senior Area Chairs and Area Chair,

Thank you for taking on additional work under such unusual circumstances. We have prepared this concise author summary to help you quickly understand how we have addressed the key concerns raised in the reviews, as well as the core contributions of our work.

**1. Addressed Reviewer L4z9’s Concerns**

Reviewer L4z9 provided an overall positive evaluation of the system’s clinical relevance but raised several constructive concerns. Below is a concise summary of how each key point has been fully addressed.

**• Novelty vs. Existing Medical Agent Systems & Medical World Model**

As requested, we expanded Related Work and added a dedicated paragraph clarifying FUAS-Agents’ distinct innovations. We also added a direct comparison with the Medical World Model, highlighting that it performs disease-trajectory simulation rather than producing executable surgical plans.

**• Memory (Top-K) & Optimizer vs. Beam/Tree Search**

The reviewer questioned the simplicity of the Top-K Memory and the lack of comparison with beam/tree search. We clarified the design rationale: Top-K is widely adopted in clinical retrieval for its efficiency and interpretability, and the Optimizer Agent performs constraint-aware LLM self-refinement rather than combinatorial search. We strengthened this with quantitative improvements shown in Appendix Table A.

**• Missing Medical LLM Baselines**

We added these baselines exactly as requested. FUAS-Agents demonstrated superior clinical reasoning and applicability in expert blind evaluation.

**• Missing Dose Prediction Abalation**

We added ablations and multiple baseline regressors (Bayes-XGB, SVR, RF), confirming that our selected Lasso-XGBoost model performs best in MAE, AUC, and R².

Together, these updates directly address all of Reviewer L4z9’s concerns and significantly strengthen clarity, methodological justification, and baseline completeness.

**2. Addressed Reviewer qWo3’s Concerns**

Reviewer qWo3 provided a strongly positive evaluation (Rating: 8) of FUAS-Agents’ innovation, empirical strength, clinical relevance, and interpretability. And raised several points regarding system stability, data generalization, safety, and potential broader applicability.

**• Stability of Reflection Mechanism**

The reviewer noted the potential risk of overfitting or instability in multi-turn reflection-based optimization. As requested, we clarified and bounded the Optimizer Agent to at most two reflection rounds, with all updates constrained by clinical guidelines and validated memory items. Any unresolved conflicts trigger mandatory clinician intervention. This evidence-guided, constraint-controlled, human-in-the-loop design ensures reliable and safe plan generation.

**• Generalization Across Institutions and Devices**

Concerns were raised about single-center MRI data and its impact on generalization. We highlighted that the Strategy Agent was fine-tuned using reports from three medical centers, and the MRI dataset already covers multiple device generations, introducing realistic heterogeneity.

**• Safety and Reproducibility**

The reviewer pointed out risks if suboptimal or inconsistent plans are generated, and potential barriers due to private data reliance. We emphasize multi-layered safety checks (guideline verification, memory retrieval, forced clinician review), ensuring unsafe plans cannot be applied clinically. To guarantee reproducibility, we have released all code, data schemas and workflows.

**• Transferability to Other Precision Therapies**

Importantly, the reviewer questioned whether FUAS-Agents could be applied to other diseases. We clarified that the system is a modular, portable paradigm. Its agent framework allows substitution of domain-specific models and knowledge bases, making it directly applicable to other precision therapies, such as radiotherapy or focused microwave ablation, thereby highlighting the broader impact and generalizability of our approach.

These clarifications and updates have been fully implemented in the revised manuscript.

---

> ### Author Response · Authors · 2025-12-03
> **Author Summary Comment（2/2）**
>
> **3. Addressed Reviewer Nfrz's Concerns**
>
> Reviewer Nfrz provided detailed feedback and raised points. We focused on these concerns that are actionable and improve the manuscript.
>
> **• Fair Strategy Generation**
>
> We clarified that treatment-strategy generation is strictly text-to-text, using only structured MRI report text, and patient background. Segmentation outputs and dose-prediction results are not used at this stage. Consequently, benchmarking against medical or general-domain LLMs under text-only conditions is fair.
>
> **• Expanded Baselines and Human Evaluation**
>
> In response, we added three strong medical LLM baselines (Baichuan2-Med, HuatuoGPT, LLaVA-Med) and conducted a multi-dimensional blind evaluation by human experts. This makes the evaluation more comprehensive and robust.
>
> **• Improved System Transparency**
>
> We enhanced the manuscript with explicit symbolic definitions of each agent and detailed module interactions. A new comprehensive data-flow figure illustrates how imaging data, clinical text, tool modules, and agents interact, resolving prior concerns about unclear module boundaries and improving overall interpretability.
>
> These changes substantially improve clarity, system interpretability, and fairness, directly addressing the reviewer’s main concerns. While the reviewer’s numerical score remained unchanged, we believe the revisions materially strengthen the manuscript.
>
> **4. Summary of core contributions**
>
> We believe the paper makes substantive and clearly defined contributions in the following areas:
>
> **(1) A clinically grounded solution to real challenges in FUAS treatment planning**
>
> FUAS is an important noninvasive therapy, yet treatment planning remains highly experience-dependent, with limited consistency, and a long learning curve for novice physicians. Our work directly targets these real-world challenges and provides a system aligned with clinical needs.
>
> **(2) A design fully aligned with real clinical processing, offering strong interpretability**
>
> FUAS-Agents disassembles and mimic the actual treatment planning processing, including image understanding and analysis, dose estimation, and strategy generation. Each agent provides transparent logic and structured outputs that match the reasoning patterns of human experts, thereby improving interpretability and safety.
>
> **(3) A combination of Specialized small models and LLM improves accuracy and robustness**
>
> By integrating MedSAM2 segmentation, XGBoost dose prediction, and a FUAS-specific RAG-enhanced LLM, the system achieves substantially higher precision in extracting key clinical elements and generating clinically coherent plans.
>
> **(4) A generalizable paradigm for multi-agent medical decision systems**
>
> Beyond a single system, our approach represents a transferable paradigm: small-model plus large-model collaboration, private-data fine-tuning, RAG-based medical knowledge augmentation, and coordinated multi-agent reasoning.
> Future domain-specific medical LLMs will inevitably rely on institution-owned private data and controlled fine-tuning to solve highly customized clinical tasks. Our system provides a concrete and scalable blueprint for this emerging direction.
>
> **(5) A Significant superiority in expert blind evaluation**
>
> In a multi-dimensional blind test conducted by experienced physicians, FUAS-Agents significantly outperformed all baseline models (including the newly added medical LLMs) in terms of clinical accuracy, completeness, and risk awareness, further validating the rationality and practical applicability of our method.
>
> **5. Conclusion**
>
> Although the rebuttal phase ended early due to system issues, we have made every effort to provide complete clarifications, add strong new baselines, and supplement architectural details. We respectfully invite you to consider the original reviews, our detailed responses, the updated manuscript, and the clarifications summarized here.
>
>
> Thank you again for your considerable time and effort under these special circumstances.
>
> Sincerely,
>
> The Authors

---

### Meta-Review · Area_Chair_NXZi · 2026-01-13

**Summary:**

The reviewers were unconvinced on the positive side, with one Reject, one marginally below the acceptance threshold, and one accept. Two of the reviewers agreed that this work requires additional effort to meet the ICLR acceptance bar. Thus, I am inclined not to accept this draft at this stage. Thank you for your effort and careful follow-up. It is an interesting work. I hope the input from the reviewers will help you further improve this work.

**Reviewer Concerns:**

Two of the reviewers are concerned about the clarity of the presentation, the evaluation, and the benchmarking protocols in this work.

**Reviewer Scores:**

N/A

---

### Decision · Program_Chairs · 2026-01-26

Reject